# IdentifiHR predicts homologous recombination deficiency in high-grade serous ovarian carcinoma using gene expression

Ashley L. Weir [1,2] ✉, Samuel C. Lee [1,2,3,4], Mengbo Li [1,2], Ahwan Pandey [5,6], Chin Wee Tan [1,2,7], Dale W. Garsed [5,6], Susan J. Ramus [8] & Nadia M. Davidson [1,2] ✉

## Abstract

**Background** Approximately half of all high-grade serous ovarian carcinomas (HGSCs) have a therapeutically targetable defect in homologous recombination (HR) DNA repair. While there are genomic and transcriptomic methods, developed for other cancers, to identify HR deficient (HRD) samples, there are no gene expression-based tools to predict HR status in HGSC specifically. We have built a HGSC-specific model to predict HR status using gene expression. **Methods** We separated The Cancer Genome Atlas (TCGA) cohort of HGSCs into training ($n = 288$) and testing ($n = 73$) sets and labelled each case as HRD or HR proficient (HRP) based on the clinical standard for classification. Using the training set, we performed differential gene expression analysis between HRD and HRP cases. The 2604 significantly differentially expressed genes were used to train a penalised logistic regression model. **Results** IdentifiHR uses the expression of 209 genes to predict HR status in HGSC. These genes preserve the genomic damage signal, capturing known regions of HR-specific copy number alteration which impact gene expression. IdentifiHR is 85% accurate in the TCGA test set and 86% accurate in an independent cohort of 99 samples, taken from primary tumours, ascites and normal fallopian tubes. Further, IdentifiHR is 84% accurate in pseudobulked single-cell HGSC sequencing from 37 patients and outperforms existing expression-based methods to predict HR status, being BRCAness, MutliscaleHRD and expHRD. **Conclusions** IdentifiHR is an accurate model to predict HR status in HGSC. It is available as an open source R package, empowering researchers to robustly classify HR status when only transcriptomic sequencing data is available.

## Plain language summary

High-grade serous ovarian cancer (HGSC) is a type of ovarian cancer with very poor outcomes. However, half of HGSCs have faulty DNA repair that can be targeted for treatment if it is identified. Existing methods look at changes in DNA that arise when repair is faulty, but do not consider which genes are actively being used, or are "expressed", by the cancer. We developed IdentifiHR, a machine learning method to predict DNA repair status using the expression of 209 genes. We tested IdentifiHR on 209 patient samples and found it correctly predicts repair status in about 85–86% of cases, performing better than existing tools on the same patient data. IdentifiHR is released as a software package for public use.

## Background

Homologous recombination (HR) is a DNA repair mechanism that is essential to the maintenance of genomic integrity. HR facilitates the repair of DNA damage that results in the stalling of replication forks and/or causes double-stranded DNA breaks and interstrand crosslinks. It is regarded as a

conservative, error-free mechanism of repair due to its reliance on the homologous DNA sequence to guide repair. HR deficiency (HRD) causes a dependency on alternate, error-prone mechanisms of DNA damage repair, which results in genomic instability and HR-specific copy number scars across fragile regions of the genome. HRD is a common molecular feature of

[1]The Walter and Eliza Hall Institute of Medical Research, Parkville, VIC, Australia. [2]Department of Medical Biology, Faculty of Medicine, Dentistry and Health Sciences, The University of Melbourne, Parkville, VIC, Australia. [3]Olivia Newton-John Cancer Research Institute, Heidelberg, VIC, Australia. [4]School of Cancer Medicine, La Trobe University, Bundoora, VIC, Australia. [5]Peter MacCallum Cancer Centre, Melbourne, VIC, Australia. [6]The Sir Peter MacCallum Department of Oncology, The University of Melbourne, Melbourne, VIC, Australia. [7]Frazer Institute, Faculty of Medicine, The University of Queensland, Woolloongabba, Brisbane, QLD, Australia. [8]School of Clinical Medicine, UNSW Medicine and Health, University of NSW Sydney, Sydney, NSW, Australia. ✉e-mail: weir.a@wehi.edu.au; davidson.n@wehi.edu.au

ovarian, breast, prostate, and pancreatic carcinomas, among others, and largely occurs due to germline and somatic mutations in HR-related genes such as BRCA1/2[1,2]. However, epigenetic events also give rise to HRD, and in some cases the molecular cause remains unknown. Notably, HRD is of great clinical importance as it imparts platinum sensitivity and can be exploited for treatment with poly-ADP-ribose polymerase inhibitors (PARPi). PARPi have already been approved in the clinical management of some ovarian and breast carcinomas, particularly when a BRCA1/2 mutation has been detected[3].

High-grade serous ovarian carcinoma (HGSC) is the most common and aggressive histotype of ovarian carcinoma, accounting for ~70% of all diagnoses. Patient outcomes are typically poor, and treatment options are limited. However, approximately 50% of HGSCs are HRD and consequently, are characterised by genomic instability and responsive to PARPi[4]. A study by The Cancer Genome Atlas (TCGA) revealed mutations to the HR genes, BRCA1 and BRCA2, occur in 12.5% and 11.5% of HGSCs, respectively, though the majority of these changes were detected in the germline (9% and 8% respectively)[5]. Together, BRCA1/2 mutations occur in 16.7%-28% of HGSCs[6–9]. Further deleterious mutations to the double-strand DNA break repair genes, RAD51C and RAD51D, have been shown to confer moderate susceptibility to HRD, and hypermethylation of BRCA1 and RAD51C promoters is identified in 14% of HGSC cases[10]. PARPi remains the only targeted therapy routinely used against HRD HGSCs[3]. It is vital that the status of HR repair can be accurately determined to ensure PARPi are used to treat HRD HGSCs, that are likely to respond, and to better understand the molecular features of HR proficient (HRP) HGSCs, which are unlikely to respond.

There are several methods to estimate HR repair status using DNA. Single-nucleotide polymorphism (SNP) array analysis of the genome has identified three patterns of copy number change and chromosomal instability likely to be caused by HRD, being telomeric allelic imbalance (TAI), large-scale transition (LST) and loss of heterozygosity (LOH)[11–13]. TAI represents the number of genomic regions with an allelic imbalance that extends into the telomere but does not cross the centromere[12]. The number of lost chromosomal regions, shorter than the entire chromosome but larger than 15 Mb, define LOH[13]. LST defines the number of breakpoints between adjacent regions (within at least 10 Mb of each other after filtering copy number variants)[11]. Taken together, these patterns give an accumulated score of genomic damage, that, when considered with germline and/or somatic mutations in HR genes, specifically BRCA1/2, represents the gold standard for the detection of HR and PARPi efficacy in clinical practice[14]. Clinical-grade assays, such as the Myriad myChoice CDx, take this HRD score into account alongside other variables, though it is accepted that this "genomic scarring" does not always reflect the underlying genomic instability or predict treatment response. HR repair can be restored through events such as reversions of BRCA1 mutations, and this is not adequately represented by the HRD score[15]. As an HRD score reflects the historic HR status and not the current functional HR status, the score cannot always be reliably used as a predictive biomarker of platinum chemotherapy or PARPi response. Further, there is evidence that HGSC patients deemed HRP by an HRD score can be responsive to PARPi, highlighting that while this is the gold standard, it does have limitations.

Alternate methods for estimating HR status from genomic signal utilise single-base substitutions (SBS), small insertions and deletions (ID), copy number change and structural variants[16–18]. In a pan-cancer context, mutational signatures, including the SBS signature 3 and ID signature 6 have been linked to HR-based DNA damage and are strongly associated with germline and somatic BRCA1/2 mutations and BRCA1 promoter methylation in HGSC, in addition to breast and pancreatic cancers[19]. Mutational signatures are not currently implemented in a clinical setting, though they represent an important insight into the molecular features that cause and/or result from HRD. Macintyre et al. built chromosomal instability (CIN) signatures for HGSC based on copy number variation (CNV) derived from shallow whole genome sequencing[16]. Pan-cancer CIN signatures have also

been developed by Drews et al. and Steele et al. derived from shallow whole genome sequencing and SNP array data[17,20]. These signatures summarise features of cancer genomes, including copy number, chromosomal breakpoints, and chromosomal segment size, and propose aetiologies through mutational and pathway-based associations. While mutational and CIN signatures have the potential to aid the classification of HR status, and reveal consequences of HRD, they are not designed or tested for the classification of a sample as being either HRD or HRP.

Machine learning algorithms have also been applied to genomic data to predict HR status, using similar input features. While these "mutational signatures" largely rely on whole-genome sequencing (WGS), SigMA, a machine learning surrogate method to detect SBS3 that was developed to use not only WGS but also whole exome sequencing (WES) and targeted gene panels[21]. In contrast, Classifier of HOmologous Recombination Deficiency (CHORD) is a pan-cancer random forest model that uses 29 mutational features, SBS, ID and structural variants, to predict the probability of being BRCA1 and BRCA2-dependent HRD, the total probability of being HRD and to provide a discrete HR status per sample[22]. Like CHORD, HRProfiler also uses mutational features to predict HR status[23]. Though distinctly, HRProfiler is a linear kernel support vector machine that requires only Six mutational features and has been trained for only breast and ovarian cancers. Other machine learning classifiers, such as HRDetect and ScarHRD, have been trained to identify HRD in breast cancer, though their application in HGSC is unverified[24,25]. As none of these tools have been trained specifically for HGSC-samples, it is difficult to assess the relevance of model features in a HGSC context. Further, there has been little benchmarking of these genomic tools or genomic signatures.

Gene expression offers insight into the functional state of cells, and RNA sequencing is a cheaper alternative to WGS. Yet, despite the widespread adoption of RNA sequencing in research and occasionally clinical contexts, there are currently no methods using gene expression dedicated to classifying HR status in HGSC specifically. Historically, a gene expression signature indicative of BRCA1/2 mutations termed "BRCAness" was considered as a proxy of HR status, though it neglected non-BRCA mediated HRD and was never implemented in clinical practice[26]. There are fewer attempts reported in the literature to classify HR status using gene expression, as opposed to DNA. MultiscaleHRD is a multinomial elastic net regression model, developed for breast cancer, that utilises the expression of 228 genes to classify samples as HRD or HRP, in addition to BRCA1- and BRCA2-like HRD[27]. It was not released as a functional model, but rather a set of gene scores to be utilised and has not been built or tested for HGSC. However, it has shown promise in classifying HR status from single cell RNA sequencing data. Recently, Lee et al. released an elastic net penalised regression model, expHRD, to predict HRD scores in a pan-cancer context using transcriptomic data[28]. They provide users with a web interface to support prediction and report superior performance in predicting high HRD scores, in comparison with the ScarHRD algorithm. Kang et al. also recently developed a lasso logistic regression model to predict HR status pan-cancer, using the expression of 5128 genes, though no working model was released, nor was a complete list of model genes, and no benchmarking was performed[29]. While methods have been developed in breast cancer and pan-cancer, a dedicated method for HGSC, the cancer with the highest rate of HRD, is needed due to the unique nature of the disease.

To address the lack of a gene expression-based classifier for HR status in HGSC, we present IdentifiHR, an elastic net penalised logistic regression model that can accurately predict the probability of a HGSC sample being HRD or HRP using the expression of 209 genes (Fig. 1). IdentifiHR is trained on 160 HRD and 128 HRP samples from the TCGA HGSC cohort, where the genome-based HRD score from matched samples is used as the ground truth. We find that the model features capture known regions of HR-specific CNV, which impact gene expression levels. Further, the probability of predictions assigned to samples by IdentifiHR preserves the continuous nature of the underlying HRD score. We test this tool in several independent cohorts of bulk RNA-sequencing, including 73 hold-out samples from

Fig. 1 | Tuning and training IdentifiHR, an elastic net penalised logistic regression model to predict HR status in HGSCs. RNA sequencing from the TCGA HGSC training cohort was filtered, and differential expression analysis was performed. Significantly differentially expressed genes (adjusted $p < 0.05$) were then transformed, scaled and used as input features to tune a penalised logistic regression model, through five-fold cross validation. After tuning L1 and L2 hyperparameters, we trained an elastic net penalised logistic regression model, IdentifiHR, which can predict HR status using the expression of only 209 genes in HGSC. Created, in part, in BioRender. Weir, A. (2025); https://BioRender.com/isms2aw.

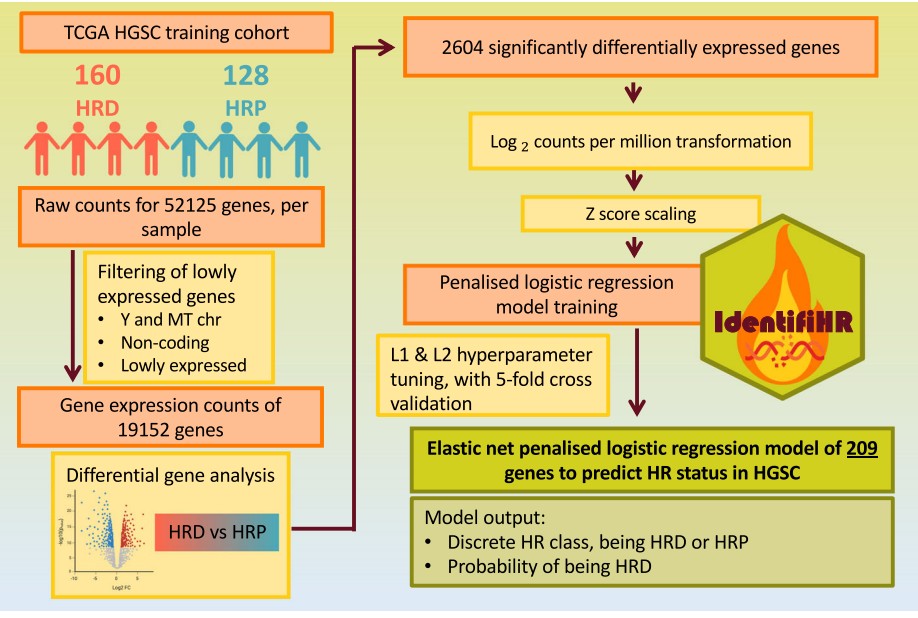

TGCA, 99 samples from the Australian Ovarian Cancer Study (AOCS), and in a pseudobulked single-cell RNA sequencing cohort of 37 patients profiled by 106 samples from the Memorial Sloan Kettering Cancer Center (MSKCC)[30]. Further, we find that IdentifiHR outperforms the existing gene expression-based methods BRCAness, MultiscaleHRD and expHRD. Our results are robust to tumour purity and read depth, and the model was accurate when tested on samples taken following autopsy, as well as those from ascites and normal fallopian tube. Our model is also accurate in predicting HR status in pseudobulked HGSC cells, suggesting the gene signature is HGSC-cell specific and not highly dependent on the tumour microenvironment. IdentifiHR is an R package and is available for download from: https://github.com/DavidsonGroup/IdentifiHR.

## Methods
### Data sources
**The TCGA cohort**. To build a gene expression classifier of HR status, we randomly separated the TCGA HGSC cohort ($n = 361$) into 80% training and 20% testing sets ($n = 288/361$ and $n = 73/361$) (Table 1 and Supplementary Data 1). Gene expression, methylation and copy-number data were all downloaded from the Genomic Data Commons (GDC; TCGA project) data portal (https://portal.gdc.cancer.gov/, https://www.cancer.gov/tcga)[5]. Illumina HiSeq 2000 RNA sequencing was downloaded as pre-processed counts for 361 patients with serous ovarian cystadenocarcinoma, also known as HGSC. Reads had been aligned to the human reference genome GRCh38/hg38 and quantified using STAR v2.7.5c, based on the GENCODE v36 annotation. SeSAMe estimated methylation beta values from the Illumina Infinium HumanMethylation27 BeadChip CpG array were downloaded for the 4 probes mapping to the *BRCA1* promoter (cg04658354, cg10893007, cg19088651, cg19531713) and were summarised as the mean beta values per sample, and for the only probe mapping the *RAD51C* promoter (cg14837411)[31]. Absolute, gene-level copy number had been called using the ASCAT2 algorithm on Affymetrix SNP6 array data, which first generates Allele-specific Copy Number Segment data with integer copy number values, from which integer Gene-Level Copy Number is derived[32].

These 361 cases also had matched clinicopathological and mutational (germline and somatic) profiling data. Clinical features including age, race and stage of disease (FIGO (International Federation of Gynaecology and Obstetrics) stage I–IV), were also retrieved. The Welch's two-sided *t*-test was applied to test for significant differences in age between training and testing sets. The mean age of individuals was not significantly different

across the training ($\bar{X}$ age = 59.9) and testing ($\bar{X}$ age = 59.5) sets (Table 1). Chi-squared testing was applied to distinguish differences in the proportion of HGSCs by FIGO stage and race, across HRD and HRP groups. The proportion of each FIGO stage and race was not significantly different across HRD and HRP cases of the training or testing set, though stage IIIC disease was most common, and most samples were provided by Caucasians (Supplementary Tables 1–2). The germline and somatic mutational status of *BRCA1/2* was obtained from The Cancer Genome Atlas Research Network analysis[5]. 51 *BRCA* mutations were detected in 48 of the 361 TCGA HGSC cases (13.3%). 30 *BRCA1* mutations were observed across 30 unique cases. The majority of these were germline (21/30 mutations), as opposed to somatic (9/30 mutations), and co-occurred with loss of heterozygosity (17/21 germline and 8/9 somatic). Similarly, 21 *BRCA2* mutations were observed across 20 HGSC cases, the majority were germline rather than somatic (15/21 and 6/21, respectively) and co-occurred with loss of heterozygosity (10/15 and 5/6, respectively).

A consensus purity estimate of each sample taken from immunohistochemistry and predictor algorithms, being ABSOLUTE, ESTIMATE and LUMP, were obtained[33]. Scores of HRD in HGSC were calculated using the ScarHRD algorithm, retrieved from the University of California, Santa Cruz Xena platform (http://xena.ucsc.edu/) and derived from WES[25,34–36]. Estimates of ploidy and absolute allelic copy-number used were estimated using ABSOLUTE[37]. The GATK4 Mutect2 pipeline and Manta had been used to call simple nucleotide variants, including both point mutations and Indels, and structural variants, respectively, from WGS of 210 of the 361 HGSCs and were also downloaded from the GDC data portal[5,38,39]. These simple and structural variant calls were used as input for CHORD to predict HR status from WGS[22].

**The International Cancer Genome Consortium (ICGC) AOCS cohort.** The AOCS cohort consisted of 81 cases represented by 99 samples collected from primary tumours before ($n = 74/99$) and after autopsy ($n = 6/99$), in addition to ascites ($n = 12/99$) and normal fallopian tube samples ($n = 7/99$), that had been profiled for HR status[40] (Table 1). Illumina HiSeq 2000 RNA sequencing was paired-end, 100 bp, and downloaded as counts from the Gene Expression Omnibus (accession: GSE209964). Downloaded counts had been pre-processed. Reads were aligned to the human reference genome GRCh37.92/hg19 and quantified using HTSeq (v0.10.0), based on the Ensembl release GRCh37.92 GTF annotation. Clinicopathological and technical variable data were available for these samples, including *BRCA1/2* mutational status. WGS was performed

**Table 1 | Training and testing cohorts used in the development and validation of IdentifiHR**

| Cohort | Cases (n) | Samples (n) | Training samples (n) | Testing samples (n) | HRD (n) | HRP (n) | Mean age (min - max) | FIGO stage (mode) | Mean tumour purity (min - max) | RNA sequencing platform | Mean read length | Median library size (min - max) | Source (PMID) |
|---|---|---|---|---|---|---|---|---|---|---|---|---|---|
| TCGA | 288 | 288 | 288 | 0 | 152 | 136 | 59.92 (34–87) | Stage IIIC | 0.86 (0.52–1.00)[a] | Illumina HiSeq 2000 | 75 bp | 60,237,474 (17,409,619–143,721,571) | 21720365 |
| TCGA | 73 | 73 | 0 | 73 | 42 | 31 | 59.45 (30–87) | Stage IIIC | 0.86 (0.60–1.00)[a] | Illumina HiSeq 2000 | 75 bp | 63,520,620 (26,171,923–134,252,345) | 21720365 |
| AOCS | 81 | 99 | 0 | 99 | 52 | 47 | 58.84 (39–78) | Stage IIIC | 0.62 (0.30–0.92)[b] | Illumina HiSeq 2000 | 100 bp | 89,817,975 (35,616,656–144,374,239) | 36456881 |
| MSKCC | 37 | 106 | 0 | 37 (pseudobulked at patient level) | 23 | 14 | 61.60 (38–81) | Stage IVB | NA (only HGSC cells examined) | Illumina HiSeq 2500 or on a NovaSeq 6000 | 28 bp/91-bp, 100 bp or 150 bp | 56,146,402 (1,094,415–325,475,953) | 36517593 |

[a]Purity taken from a consensus purity estimate published by Aran et al. (PMID: 26848121).
[b]Purity estimated from FACETS (v0.6.1) by Garsed et al. (PMID: 36456881).
AOCS Australian Ovarian Cancer Study, FIGO International Federation of Gynecology and Obstetrics, HRD Homologous Recombination Deficient, HRP Homologous Recombination Proficient, MSKCC Memorial Sloan Kettering Cancer Center, TCGA The Cancer Genome Atlas.

using 100 bp paired-end reads on the Illumina HiSeq 2000 platform to an average 40-fold base coverage for germline DNA and 52-fold coverage for tumour DNA[41]. Tumour purity and ploidy were estimated from WGS data using FACETS[42]. HR status was determined from WGS data using CHORD and scarHRD to derive an HRDsum score, as previously described[14]. As HRDsum was developed for SNP array data, not WGS, a Receiver Operating Curve (ROC)-based threshold adjustment was performed against discrete CHORD labels, to binarise the HRD score into discrete HR statuses.

**The MSKCC cohort.** The MSKCC collected 106 samples from 37 newly diagnosed, treatment-naïve patients (Table 1)[30]. Both tumour and ascites were collected from primary and metastatic sites, including bilateral adnexa, omentum, pelvic peritoneum, bilateral upper quadrants, and bowel. Of the 37 patients, 27 had adnexal, 14 had ascitic and 36 had non-adnexal tissue samples. As described by Vazquez-Garcia et al., sequencing libraries were prepared using the Chromium Single-Cell 3′ Reagent kit v3 (10x Genomics, PN 1000075). Single-cell RNA sequencing was performed on Illumina HiSeq 2500 in rapid mode or on a NovaSeq 6000, in a 28-bp/91-bp, 100-bp/ 100-bp or 150-bp/150-bp paired-end manner. Reads were processed and filtered to counts, and cell types were annotated as previously described[30]. Raw counts for only HGSC cells were downloaded as a Seurat v4 RDS object from Synapse (SynID: syn51091849). Counts were pseudobulked by summing those for all cells of the same sample to the gene level. Clinicopathological variable data was published for these samples also, including a consensus HR status at the patient level determined by authors of the source publication. The consensus HR status was based on combinations of the Myriad Genetics 'myChoice CDx' assay, the HRDetect classifier and the CHORD algorithm (not all were available for each patient), in addition to the presence of *BRCA1/2* mutations, whereby a mutated sample would be labelled as HRD, regardless of the assay-defined label.

**Ethical considerations for data sources.** Ethical approval and/or patient consent were not required for this study, as all datasets utilised were publicly available. All data used in this study comply with ethical regulations. Approval and informed consent for sample collection, processing and the subsequent sharing of data were already obtained by the groups of the original studies (The Cancer Genome Atlas, or TCGA, the Australian Ovarian Cancer Study group, or AOCS group, and Memorial Sloan Kettering Cancer Center, or MSKCC). The study was conducted in accordance with the Declaration of Helsinki.

**Determining the "true" HR status**
HRD scores are unweighted sums of three genomic features found to be indicative of disrupted HR: TAI, LOH and LST. As previously reported and frequently used in clinical practice, a HRD score of ≥42 was used as a cut-off for deficiency, with <42 indicating HR proficiency, however, it should be acknowledged that HRD would ideally be viewed as a continuous scale[43]. Where a cohort had utilised additional sources for HR status classification, they have been listed above and labels provided by the authors of the original manuscripts were used.

As a potential cause of HRD, the hypermethylation of promoter regions of *BRCA1* and *RAD51C* was examined to identify potential gene silencing. The mean methylation beta value across the four probes mapping to the *BRCA1* promoter was calculated and assessed alongside *BRCA1* gene expression to reveal that all cases, in the TCGA training and testing cohort, with a beta value of >0.15 had been defined as HRD by the HRD score (Supplementary Fig. 1A). As only one probe mapped to *RAD51C*, the exact beta values were assessed alongside expression and for all TCGA cases. All cases with a beta value of > 0.15 had been defined as HRD by the HRD score (Supplementary Fig. 1B).

**Training IdentifiHR: feature selection and transformation**
Using the TCGA training set only, we manually removed non-coding genes and those on mitochondrial and Y chromosomes. Differential expression

(DE) analysis was used to reduce the feature space further by identifying genes likely to be relevant in the prediction of HR status. Genes with low counts across all libraries were filtered using default settings of the filterByExpr function (edgeR package, v4.2.1) - at least 10 counts per million (CPM) per sample, in at least 70% of the minimum design group size, being 128 samples (derived from the group with the smallest sample size). This resulted in 18,761 of the original 52,125 genes being retained. Trimmed mean of M-values normalisation was performed to correct composition biases between libraries (using the calcNormFactors function, limma package, v3.60.4). Testing for DE genes was undertaken using limma-voom[44]. HR status, as defined above, but no other covariates were modelled in the design matrix. A voom transformation was applied to the data to adjust for library size differences. A linear model was fit to the data, and a contrast of HRD–HRP was applied to the fit. Robust empirical Bayes was employed to estimate moderated $t$-statistics and associated $p$ values, for genes. DE significance was defined by a Benjamini-Hochberg adjusted $p < 0.05$. DE genes were visualised, and gene ontology enrichment analysis was performed (using goana, limma package). The chromosomal position of DE genes was also evaluated, and differences between the proportion of significantly up- and downregulated genes on each chromosome arm compared to the proportion of significantly up or downregulated genes on all other chromosome arms were assessed through Chi-squared testing with Yates' continuity correction, whereby significance was corrected by the Bonferroni method and defined at an adjusted $p$ value < 0.05. The 2604 genes with significant differential expression were retained as potential model features. A vector of Ensembl identifiers for these DE genes was then stored to aid the filtering of new samples when being pre-processed for input into the model.

### Training identifiHR: feature standardisation

The counts of the remaining 2604 genes were normalised for library size differences, using log$_2$CPM, with the library size calculated using only the subset of 2604 genes. Each gene was independently transformed to relative expression values using a z score transformation of the log$_2$CPM values. This transformation ensures that the weights given in the resultant model can be interpreted equally across genes, independent of mean abundance levels. The mean and standard deviation of log$_2$CPM for each gene were saved as vectors and stored to facilitate z score calculation of new samples when using the model.

### Training identifiHR: hyperparameter tuning, cross-validation, and model training

To predict HR status from the z score standardised abundances of DE genes, a penalised logistic regression model was built using the R package glmnet (v4.1-8)[45]. To ensure our model was robust and generalisable, we fit a model with both L1 and L2 regularisation, controlling their relative contribution by optimising the elastic net mixing parameter alpha. Through 5-fold cross-validation, we performed a grid search of alpha values ranging from 0 (ridge penalisation) to 1 (lasso penalisation) in intervals of 0.01 and the corresponding optimal minimum value for the lambda hyperparameter, taken from 100 values generated within the function cv.glmnet. Following the one-standard error rule, the alpha value with the smallest standard deviation across the five folds, within one standard deviation of the highest area under the receiver operating curve (AUC) for any alpha value, was selected. These optimal values were then used to train an elastic net logistic regression model on the entire training cohort that predicts the probability of a new sample's HR status being HRP. To report the probability that a sample is HRD, we invert this and subtract the raw probability from one. The features with non-zero beta coefficients were inspected; potential pathway enrichment in these features was assessed through gene ontology by goana (limma package), and the genomic location of features was considered. Potential co-expression between weighted genes and *BRCA1/2* was assessed using Pearson's correlation coefficient, comparing the log$_2$CPM of *BRCA1* or *2* against the log$_2$CPM of each model gene.

### Testing identifiHR

The resulting predictive tool, IdentifiHR, was then tested on independent cohorts. A hold-out subset of the TCGA was examined, alongside an independent cohort from the AOCS. In addition to these bulk RNA sequencing cohorts, IdentifiHR was tested on an independent cohort of pseudobulked HGSC cells from MSKCC. In all test cohorts, genes were subset down to the 2604 required by IdentifiHR, transformed using log$_2$CPM and scaled into a z score. Where genes were missing from cohorts, the count values were set to zero in all samples. IdentifiHR was then used to predict HR status. Testing metrics including the AUC, accuracy, precision, recall and the misclassification error were calculated. Principal component analysis (PCA) of model genes in these test samples was performed for visualisation. Where possible, the correlation between the HRD score and the probability of HR status was also assessed. As the truth labels given to test samples were based on genomic scar patterns, those labelled "incorrectly" by IdentifiHR were manually interrogated.

The probability threshold, at which IdentifiHR predictions are defined as "HRD", was optimised in the TCGA testing cohort by iterating through probability values from 0 to 1, in intervals of 0.01, and taking the threshold that maximised accuracy. The AOCS testing cohort provided independent assessment of this adjusted threshold.

### Benchmarking against gene expression methods of HR classification

To our knowledge, there are currently no other gene expression-based classifiers of HR status, built in and for HGSC. Accordingly, we have compared IdentifiHR's predictions against the BRCAness and the MultiscaleHRD transcriptional signatures, both of which were developed for breast cancer, and against the pan-cancer predictive model, expHRD[27,28,46,47]. HR status predictions were compared in the TCGA testing set. Metrics including accuracy, recall, precision, and the misclassification error were assessed based on discrete HR status predictions.

**BRCAness**. No recommendation was provided for how to infer HR status from the BRCAness transcriptional signature. Raw expression counts of the TCGA testing set were transformed using log$_2$CPM and subset down to only the 40 genes of the BRCAness signature. These transformed genes were scaled into a z score. *K* means clustering, with $k = 2$ centroids, was performed alongside dimensionality reduction by PCA to assign and visualise discrete HR classes. Assessing HR status through the BRCAness gene signature did not reveal distinct clusters of HGSCs (Supplementary Fig. 2A). Low overlap was observed between the HR status of samples and classification via k-means clustering (Adjusted Rand Index = 0.001). Cluster 1 was deemed HRP-like and cluster 2 HRD-like based on the contribution of HR genes, specifically *BRCA1/2*, whereby low expression was deemed suggestive of HRD (Supplementary Fig. 2B–D).

**MultiscaleHRD**. Gene expression counts of the TCGA testing cohort were subset down to only the 228 genes of the MultiscaleHRD model, as no tool was released to predict HR status. They were then transformed using log$_2$CPM and scaled into a z score, as recommended in the paper by the authors, and the Pearson's correlation coefficient between these gene abundances and model centroids, accessed at https://github.com/secrierlab/MultiscaleHRD was calculated for each sample. Where the correlation with the HRD model was positive, the sample was classed as HRD, and where negative, as HRP. MultiscaleHRD also produces a *BRCA1/2*-like classification, in addition to a broad HR status prediction, though no testing was performed against the *BRCA1/2*-like model output, as these groups were not directly comparable to our model's output.

**ExpHRD**. We assessed expHRD in our TCGA testing cohort and also the 99 samples of the AOCS cohort. As recommended by expHRD authors, we normalised raw RNA sequencing counts in both cohorts using the DESeq2 R package and utilised the web interface for the prediction of

HRD score, with associated 95% confidence intervals. As expHRD does not offer discrete HR status classes, we considered samples with a "predicted_HRD" <42 as HRP and those ≥42 as HRD, in accordance with the clinical gold standard. However, we also determined the optimal threshold for HR status classification in this model, iterating through predicted HRD score values ranging from 1 to 114 (being the highest HRD score predicted by expHRD in the TCGA testing cohort) to determine a cut-off that would maximise the model's accuracy. This optimal expHRD score threshold was then used to examine model accuracy in both the TCGA and AOCS testing cohorts.

### Statistics and reproducibility
All statistical analyses and modelling, including model training and testing, were carried out in R v4.2.1, and can be found at https://github.com/DavidsonGroup/IdentifiHR.

## Results
### Differential gene expression detects regions of known copy number variation associated with HRD in HGSC samples
52,125 genes were assessed in the TCGA training cohort ($n = 288$). 2604 DE genes were identified when significance was defined by an adjusted $p$ value < 0.05 (Supplementary Data 2). 1315 genes were downregulated, and 1289 genes were upregulated in HRD samples (Fig. 2A). To confirm whether the DE genes captured the genomic signature of HRD, we performed PCA on gene expression before and after subsetting to the DE genes (Fig. 2B, C). PCA of the DE gene expression separated samples as HRP and HRD in the first dimension and captured a gradual transition from low to high HRD score (Fig. 2B, C and Supplementary Fig. 3A, B).

Amongst the DE genes, *BRCA1* was the fifth most significantly downregulated in the HRD group (ranked by adjusted $p$-value) (Supplementary Data 2). As *BRCA1* is known to be important in HR repair and is commonly silenced through methylation as a mechanism of HRD, this result supports the biological relevance of the DE signal. Gene ontology analysis revealed that DE genes were significantly enriched in gene sets associated with nitrogen metabolism, oxidative damage, proteasomal degradation and DNA damage (Supplementary Fig. 3C and Supplementary Data 3). As HGSC is characterised by structural alterations of the genome, the chromosomal locations of DE genes were considered. The proportion of up- and downregulated genes differed significantly across the p and q arms of some chromosomes (Fig. 2D). Regions enriched for up- and down-regulated DE genes were found across chromosomes (Fig. 2E). For example, DE genes on chromosome 8 and X were enriched for upregulation, whereas chromosomes 19 and 20 had largely downregulated DE genes. Many of the top upregulated DE genes were present on chromosome 8, in a concentrated location at the end of the q arm (8q24.2–8q24.3), such as the most significant DE gene, *C8orf33* (Fig. 2E). Copy number amplifications in the 8q24.2 region were recently identified as being significantly enriched in HRD HGSCs and were suggested to be predictive of HR status[48]. Further, HRD HGSCs were enriched for 5q13.2 deletions, and HR proficient HGSCs were enriched for 19q12 amplifications; these CNVs were also found to be enriched in HRD and HRP cases of other cancers, supporting their relevance to the DNA repair mechanism[48]. Consistent with these known CNVs in DNA, we observed a notable incidence of downregulated DE genes at chromosome 5q13.2 and 19q12 in RNA (Fig. 2E). A moderate, positive correlation was observed between gene-level absolute copy number and significant DE genes in these three regions, in both HRD and HRP HGSCs (Supplementary Fig. 4A–C). Further, summarising the mean gene-level copy number and mean expression of these genes revealed clear distinctions between HRD and HRP cases at 8q24 and 5q13, though this was less pronounced at 19q12 (Supplementary Fig. 4 D). Several other regions of consistent up or downregulation of gene expression allude to other HR status-specific CNVs that remain uncharacterised. These results show that CNVs associated with HRD can alter gene expression in HGSC, motivating the development of a gene-expression-based classifier.

### IdentifiHR is an elastic net penalised logistic regression model based on the expression of 209 protein coding genes
Using the 2604 significantly DE genes as features, we have trained an elastic net penalised logistic regression model, IdentifiHR, to predict HR status in HGSC. We have packaged the pretrained model as an R function in the package, IdentifiHR. The mean AUC across the five folds of the training cohort was 0.86, and the mean accuracy was 0.78. The optimal alpha value, being that with the smallest standard deviation across the five-folds, within 1 standard deviation of the highest AUC for any alpha value, was determined to be 0.53 (Supplementary Fig. 5A). Of the 2604 input genes, 209 had non-zero weights in the final IdentifiHR model following elastic net penalisation (Supplementary Data 4). The beta coefficients of these model genes contributed to the classification of HR status symmetrically and were represented by a wide range of $\log_2$ fold-changes in the initial DE analysis used for feature selection (Supplementary Fig. 5B, C). Assessing the beta coefficients of the 209 model genes across chromosomal location demonstrated that IdentifiHR was capturing genomic CNV signals, with genes in regions of known HR-specific CNV being selected as IdentifiHR features (Supplementary Fig. 5D). This is evidenced by the clustering of positive beta coefficients at 8q24.2 and of negative beta coefficients at 5q13.2, though no genes within 19q12 were included in the model (Supplementary Fig. 6A–H). Novel regions, such as the end of chromosomes 15 and 17, also contributed positively to a sample being classed as HRD, though do not have known HR associations, and were found to be enriched for upregulated genes in the DE analysis (Fig. 2E).

Only one gene canonically associated with HR repair was included in the model, being *FANCI*. While *BRCA1/2* were significantly DE during feature selection, neither was weighted in the final model. As this absence could potentially be explained by weighted model genes being co-expressed with *BRCA1/2*, this association was assessed. Genes were weakly to moderately correlated with *BRCA1* and *2*, though no individual model genes were strongly correlated with the expression of either gene (Supplementary Data 4). A significant, linear association was however, observed between the beta coefficient for each model gene and its matched correlation coefficient with *BRCA1*, but not *BRCA2*, expression (Supplementary Fig. 5E, F). This is notable as it suggests that while an individual gene may not be strongly co-expressed with *BRCA1/2*, the model is associated with the signature of *BRCA1* expression. PCA of selected genes showed HRD status as the leading source of variability in the TCGA training and testing sets, in addition to the AOCS testing cohort (Supplementary Fig. 7A–F).

### IdentifiHR can accurately predict HR status in independent TCGA HGSC samples
IdentifiHR was used to predict HR status in a held-out subset of the TCGA cohort. Notably, the probability of being HRD assigned by IdentifiHR was correlated with the HRD score of the sample ($R = 0.65$) (Fig. 3A). The truth labels, taken from the HRD score, suggested 47% of samples were HRD and 53% were HRP ($n = 34$ and 39/73, respectively) (Supplementary Data 5). 62 of the 73 samples were predicted correctly by IdentifiHR, meaning they were given the same discrete HR class as their truth labels (AUC = 0.86, accuracy = 85%) (Fig. 3B and Supplementary Data 5). The model detected a low number of false positives, given by a precision of 83%, and of false negatives, given by a recall of 85%. The misclassification error was accordingly low, at 15%. The 6 samples incorrectly predicted by IdentifiHR to be HRD were assigned a wide range of probabilities, whereas 4 of the 5 samples incorrectly predicted to be HRP all had low probabilities assigned by IdentifiHR ($P < 0.026$). Notably, these samples all had HRD scores ranging from 44 to 55, being close to the clinical cut-off for HR status. The mean consensus purity estimate of cases with correctly predicted labels was 86.2% (minimum purity 59.9%, maximum purity 99.9%) and was not significantly difference from that of incorrectly predicted cases at 84.7% (minimum purity 66.6%, maximum purity 97.8%) (Welch's two-sided $t$-test).

To check the robustness of IdentifiHR with respect to how truth is defined, CHORD predictions of HR status were also assessed in the 43 samples of the TCGA testing cohort ($n = 73$), with WGS available. Across

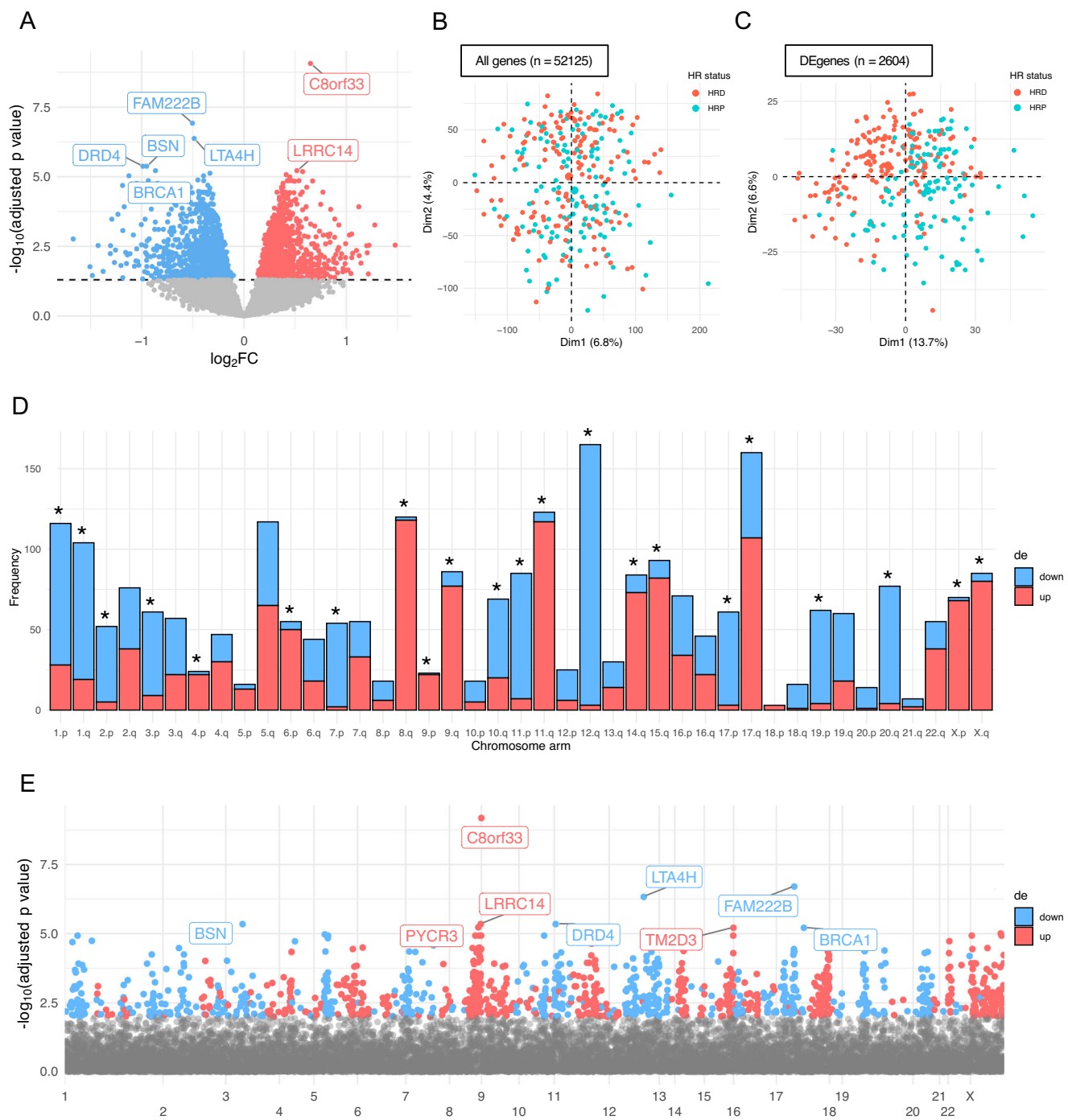

**Fig. 2 | 2604 genes were differentially expressed between HR deficient and proficient HGSCs ($n = 288$), where adjusted $p < 0.05$. A** Volcano plot showing the relationship between the $\log_2$ fold-change ($\log_2$FC) and significance, using the limma two-sided empirical Bayes method. Grey points give each gene assessed, those highlighted in red are significantly upregulated in HRD and those in blue are significantly downregulated in HRD. The dashed horizontal line indicates the cut-off for significance, being the Benjamini-Hochberg adjusted $p$ value < 0.05. The top 8 DE genes are labelled. PCA plots of the $z$ score of $\log_2$ CPM of **B** all genes ($n = 52125$) and **C** DE genes ($n = 2604$). Each point represents a TCGA training sample coloured by HR status (HRD defined as having a HRD score of ≥42). **D** Frequency of up (red) and downregulated (blue) DE genes by chromosomal arm. Asterisks indicate if the proportion of up and downregulated DE genes on each chromosome arm is significantly different to the global proportion (two-sided Chi-Squared test, Bonferroni adjusted $p < 0.05$). **E** The location of up (red) and downregulated (blue) DE genes across each chromosome. Those assessed but not significantly differentially expressed are given in grey.

both the TCGA training and testing cohort, the CHORD probability and matched HRD score were positively correlated ($R = 0.71$, $p$ value < 2.2 × 10−16) and 84.2% concordant (Supplementary Fig. 8A). In only the TCGA test cohort, these labels were 86% concordant (Supplementary Fig. 8B). CHORD labels were 79% concordant with IdentifiHR predictions in the 43 samples of the TCGA testing cohort with WGS, though some variation in the probabilities predicted by each model was observed (Supplementary Fig. 8C–F).

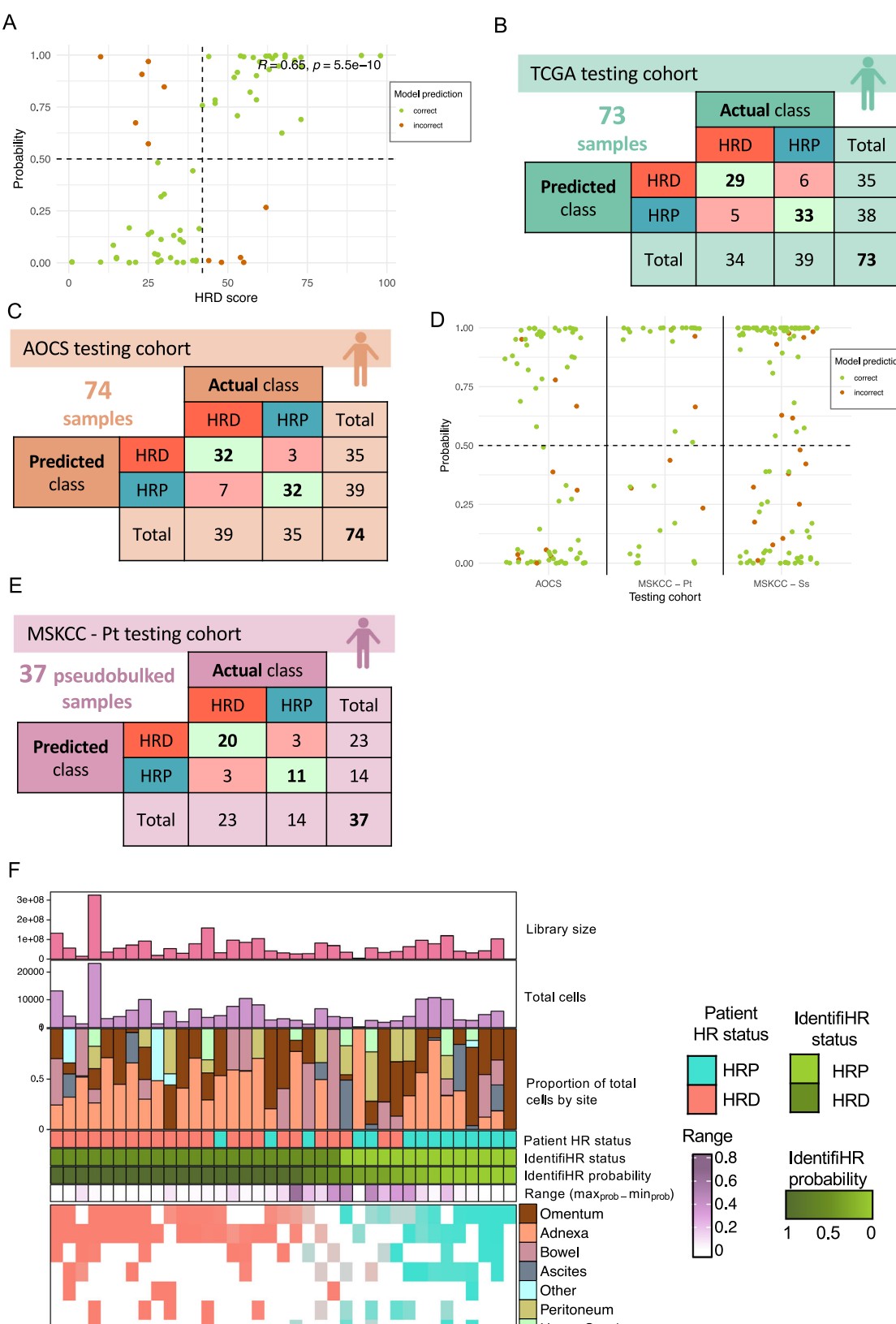

## IdentifiHR can accurately predict HR status in independent AOCS samples

The AOCS cohort of 99 samples was of particular interest in testing as it contained samples not just taken from primary HGSCs, but also included samples collected after autopsy ($n = 6$), from ascites ($n = 12$), representing a liquid biopsy alternative for HGSC, and from normal fallopian tube ($n = 7$), being the tissue of origin for HGSC. Moreover, as these samples came from a different cohort from TCGA, testing the performance of the IdentifiHR model in this cohort serves as an important validation of the model's generalisability. HR status

**Fig. 3 | IdentifiHR can accurately predict HR status in the TCGA and AOCS testing cohorts. A** The probability of being either HRP ($P < 0.5$) or HRD ($P \geq 0.5$) as predicted by IdentifiHR of each TCGA testing sample ($n = 73$) against the ground truth, HRD score. The dashed vertical line at HRD score = 42 indicates the clinical cut-off for HR status classification, and dashed horizontal line indicates the model's predictive cut-off. The Pearson correlation coefficient of a linear fit to the points (R) with the associated $p$ value is also given. **B** Confusion matrix of IdentifiHR predictions of HR status, against the "true" HR status, as above, in the TCGA testing cohort. **C** Confusion matrix of IdentifiHR predictions for the AOCS cohort ($n = 74$ primary tumour samples). **D** Predicted IdentifiHR HRD probabilities for the testing cohorts, AOCS ($n = 74$ primary tumour samples), MSKCC - Pt (patient-level pseudobulked, $n = 37$ patients) and MSKCC – Ss (site-level pseudobulked, $n = 106$ samples), with samples colored by whether the model correctly or incorrectly predicted the HR status. **E** Confusion matrix of IdentifiHR predictions for the MSKCC - Pt cohort ($n = 37$ samples pseudobulked at the patient level). **F** The IdentifiHR predicted HR status and probability for each patient (columns) in the MSKCC - Pt testing cohort, sorted by probability. Also shown is the library size, the total number of cells per sample, the proportion of the cells derived from distinct tissue types and the true HR status of the patient. IdentifiHR's predicted probabilities for HR status across site samples (MSKCC – Ss) of the MSKCC - Pt cohort, in addition to the range of probabilities predicted by patient (being the maximum probability in any sampled site minus the minimum). Created, in part, in BioRender. Weir, A. (2025). https://BioRender.com/isms2aw.

considered to be ground truth was taken from Garsed et al. and differed from that calculated for TCGA samples as a CHORD score was used, though the matched HRD score derived from HRDsum was also considered and the discrete HR status threshold of these scores was optimised (supplementary 0ure 9A, B)[40]. Using CHORD, 48% of samples were expected to be HRD and 53% to be HRP ($n = 47$ and 52/99, respectively) (Fig. 3C and Supplementary Data 6). Alternatively, HRD and HRP were each at 50%, according to the HRD score, and these genomic labels were concordant in 81% of the test cohort (Supplementary Fig. 9B). IdentifiHR indicated that 56% were HRD, and 44% were HRP ($n = 55$ and 44/99, respectively) (Supplementary Data 6). The AUC in this cohort was 0.91, and the accuracy was 86% using CHORD labels (Supplementary Fig. 9C). IdentifiHR detected a low number of false positives, given by a precision of 80%, and of false negatives, given by a recall of 94%. Congruently, the misclassification error was 14.1%. A significant, strong positive correlation was observed between the IdentifiHR probability and the CHORD probability and HRD score, respectively ($R = 0.79$, $p$ value $< 2.2 \times 10^{-16}$ and $R = 0.7$, $p$ value $= 9.7 \times 10^{-15}$) (Supplementary Fig. 9C, D). IdentifiHR perfectly predicted the HR status of all samples collected after autopsy, irrespective of if they were sequenced from whole tumour tissue or enriched by microdissection. Further, all samples of normal fallopian tube tissue were correctly predicted to be HRP ($n = 7/99$). Some discordance was observed between CHORD and HRDsum in the 12 ascites samples, with CHORD suggesting 6 to be HRD and 6 to be HRP, while the HRD score suggested 9 were HRD, with 2 of the 3 predicted to be HRP agreeing with CHORD. IdentifiHR predicted 2 of the 6 HRP samples, by CHORD, concordantly, though suggested the remaining samples were HRD. The 10 samples that were incorrectly labelled from the primary tumour, collected before autopsy, were derived from 10 distinct cases (Supplementary Fig. 9E). Of these, 3 had probabilities predicted by IdentifiHR within the range 0.25 to 0.75, and therefore being close to the cut-off of 0.5 (Fig. 3D). They had a mean tumour purity estimate of 67% (minimum purity 52%, maximum purity 90%), which was slightly higher than the correctly labelled samples which had a mean tumour purity estimate of 61% (minimum purity 30%, maximum purity 92%). Overall, there was not a significant difference in tumour purity when comparing correctly labelled cases to those incorrectly labelled (Kruskal–Wallis chi-squared = 1.0923, df = 1, $p$ value = 0.296). Of the 3 samples that were expected to be HRD by CHORD, but were predicted to be HRP, one had *BRCA1*-mediated HRD defined by a deletion event, and the other two were suggested to have *BRCA2*-mediated HRD and both harboured mutations in the HR gene, *BRIP1* (Supplementary Fig. 9E)[40]. They were the only samples in this testing cohort with documented *BRIP1* mutations. All other samples with non-*BRCA1/2* mediated HRD were correctly labelled by IdentifiHR. Of the 7 samples labelled as HRP but predicted to be HRD by our model, one had a frameshift deletion in *BRCA1* and an HRDsum score of 63, and therefore may have been truly HRD, despite the CHORD prediction. No others had documented mutations in any HR genes, though this does not exclude the presence of epigenetic modifications or structural variants that could have resulted in HRD. Further investigation is needed to clarify concordant HR status predictions by these three methods and the

incorrect prediction of HR status in these samples (Supplementary Fig. 9F).

The probability threshold for the TCGA testing cohort, which maximised accuracy, was found to be $P = 0.58$. This optimised thresholding increased the accuracy in the TCGA testing cohort from 85% to 86%, compared to the default $P = 0.50$ (Supplementary Fig. 10A). However, applying this optimised threshold to the AOCS cohort resulted in a reduction in accuracy, from 86% to 85% by CHORD and HRDsum (Supplementary Fig. 10B, C). In both cohorts, the change resulted in the re-classification of only a single sample. Given this, $P = 0.50$ is used in our package as the default; however, users are provided with the probability of a sample being HRD in the IdentifiHR output and can customise this threshold to re-define discrete HR status, based on the context of their use and the associated risk tolerance.

## IdentifiHR can accurately predict HR status in pseudobulked HGSC cells of the MSKCC

HGSC is a heterogeneous disease, and HR status can vary within and between metastatic sites of a patient[15]. This is clinically relevant as polyclonality hampers targeted treatment of HRD cells by PARPi and imparts opportunities for treatment resistance. Single-cell RNA sequencing can clarify this heterogeneity to investigate HR status subclonally. We assessed the performance of IdentifiHR in a single cell HGSC cohort to determine its potential application and ascertain if the gene expression signal used to predict HR status by the model is HGSC-specific or influenced by tumour microenvironment. Single cell RNA sequencing was performed by the MSKCC on 106 samples taken from 37 patients. 62% of the patients were expected to have HRD HGSC compared to 38% expected to be HRP ($n = 23$ and 14/37, respectively) (Supplementary Data 7). Cell types had previously been annotated by Vázquez-García et al.[30]. Excluding immune and stromal cell types, the raw gene expression counts for the 198,022 cells annotated as HGSC were pseudobulked to the patient level irrespective of their sample site. The read depth of these single-cell RNA sequencing samples following pseudobulking was comparable to the TCGA and AOCS testing cohorts, with a median library size of ~56 million reads (range = ~1 million−325 million reads) (Table 1). Genes were then subset to those required by IdentifiHR (median cells per pseudobulked patient = 3998, range = 60 to 23,097 cells). As 8 of the 209 genes (3.8%) weighted for IdentifiHR predictions were missing, their count value was set to 0. Despite the missingness, IdentifiHR had an AUC of 0.92 and an accuracy of 84% in the 37 pseudobulked samples, and both the precision and recall were 87% (Fig. 3E, F). This result demonstrates that the model is robust to different gene reference annotations. Classes were balanced across the 6 samples that were predicted incorrectly by IdentifiHR, with 3/6 being incorrectly labelled as "HRD" and the remaining 3/6 incorrectly labelled as "HRP", and majority of these samples were assigned a probability close to 0.5 (Fig. 3D). This result establishes that IdentifiHR uses a HGSC-cell specific signal to make HR status predictions, not a signal convoluted by the tumour microenvironment. It also demonstrates that the model can be applied to single-cell data, which is intrinsically different to bulk RNA sequencing as it has 5' or 3' biases and provides another example of IdentifiHR accurately predicting HR status in an independent cohort.

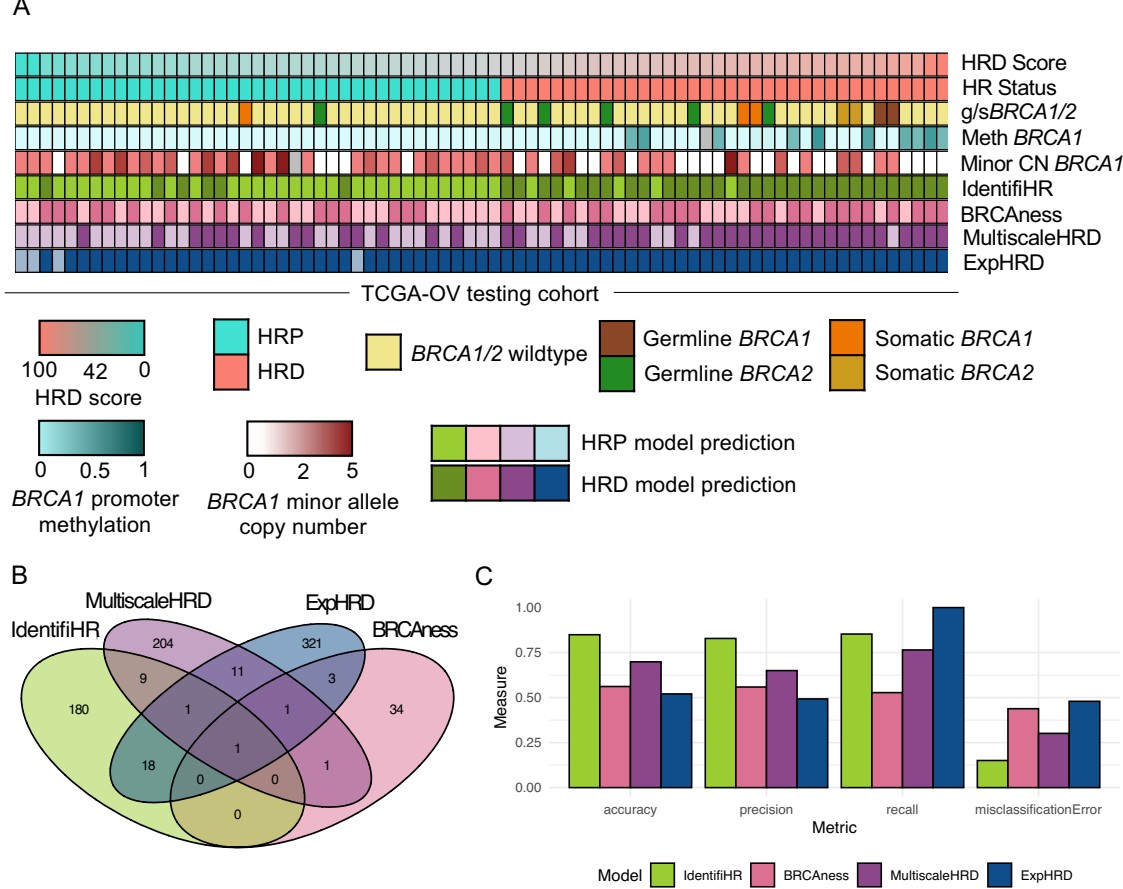

**Fig. 4 | IdentifiHR outperforms predictive tools of HR status that use gene expression, BRCAness, multiscaleHRD and ExpHRD, in the TCGA HGSC testing cohort. A** HRD predictions for each sample (columns) of the TCGA testing cohort. Samples are sorted and colored by HRD score and HR status (top two tracks). The presence or absence of germline (g*BRCA1/2*) or somatic (s*BRCA1/2*) mutations in *BRCA1/2* are annotated, with the methylation beta values for the *BRCA1* promoter ("Meth *BRCA1*"), minor allele copy number of *BRCA1* ("Minor CN *BRCA1*") and the HR status as predicted by IdentifiHR, BRCAness, MultiscaleHRD and ExpHRD are shown. Where data was not available for a case, bars are given as grey. **B** The overlap between the genes used in each of the four gene expression models assessed. **C** The performance of the predictive tools for HR status evaluated through the metrics, accuracy, precision, recall and the misclassification error.

While accurate HR status labels were not present at the MSKCC sample level, they could be inferred from the HR status of each patient. HGSC cell counts were pseudobulked at both the patient level and the sample site level, being adnexa, ascites, bowel, omentum, peritoneum, or upper quadrant. The proportion of cells derived from each tissue type differed between patients (Fig. 3F). IdentifiHR had an accuracy of 85% in these 106 samples (median cells per pseudobulked sample = 1346.5, range = 2–9613 cells), and provided predictions that were consistent with the broader patient HR status in all tissue types (Fig. 3F and Supplementary Data 8–9). Where a sample's HR status predicted by IdentifiHR was not concordant with the genomic HRD score and associated status of the patient, meaning the predicted status was "incorrect", the probability tended to be around 0.5 (Fig. 3D). As expected, the read depth of these single-cell RNA sequencing samples was low compared to the TCGA and AOCS testing cohorts, with a median library size of ~19 million reads (range = ~0.09–107 million reads) (Supplementary Data 8). While the 14 samples collected from ascites had the lowest median cell count of any site, at only 193 cells per pseudobulked sample (range = 2 to 1876 cells), 12 of their HR statuses were accurately predicted by IdentifiHR (accuracy = 86%). Further, 8 of the 106 samples across all sites had less than 100 cells, and yet IdentifiHR correctly predicted all of their HR statuses, with 5 samples being HRP and 3 being HRD. These results demonstrate that despite having much lower library sizes and in some cases, only a few cells contributing to the final gene counts, IdentifiHR can be used to accurately predict HR status in single-cell RNA sequencing.

## IdentifiHR outperforms BRCAness, multiscaleHRD and expHRD

Samples of the TCGA testing cohort were assessed by IdentifiHR, the BRCAness and MultiscaleHRD transcriptional signatures, and the expHRD pan-cancer model, to yield discrete predictions of HR status (Fig. 4A). There was little overlap in the feature genes used for predicting HR status between each of these methods, though 20 genes did overlap between IdentifiHR and expHRD (Fig. 4B). One common gene, *FANCI*, was used in all four methods and has a known association with HR (Fig. 4B). The relative ability to separate HRD and HRP samples was also observed qualitatively when gene expression was subset to the differing features of each signature, normalised, scaled and visualised with PCA (Supplementary Fig. 11A–L).

IdentifiHR had the highest accuracy of any method examined in the TCGA testing cohort, with the HR status of 62 out of 73 samples being correctly predicted (Fig. 4C). This was followed by MultiscaleHRD with 51, BRCAness with 41 and expHRD with 38 of the 73 samples having correctly predicted HR statuses (Fig. 4C). There was only one sample that was incorrectly labelled by all four methods. BRCAness and MultiscaleHRD incorrectly labelled a sample each with a germline *BRCA1* mutation. MultiscaleHRD further found 3 of the 6 cases with germline *BRCA2* mutations to be HRP, though one of these did have a HRD score < 42 and was also deemed HRP by IdentifiHR. Further, one case with a somatic *BRCA1* mutation was predicted to be HRP by BRCAness, though had an HRD score > 42. All cases with any potential hypermethylation of the *BRCA1* promoter were labelled as HRD by IdentifiHR, MultiscaleHRD and expHRD, though not BRCAness. However, 7 cases with a loss in

heterozygosity at *BRCA1* were HRP, as defined by their HRD score, and 6 of the 7 were labelled as HRD by the BRCAness signature. IdentifiHR predicted 11 of the 13 cases with *BRCA1/2* variants to be HRD. Of the 2 cases labelled as HRP with variants, one had a benign nonsense germline mutation in *BRCA2* (p.K3326*)[49,50]. The other case harboured a somatic *BRCA1* missense mutation (*p.C47W*) of uncertain functional significance, accompanied by a loss of heterozygosity, raising the possibility of biallelic inactivation. However, the low HRD score of 27 suggests the variant was unlikely to be pathogenic and that the sample was relatively HR proficient.

As an HRD score is a continuous variable, we also examined whether the samples predicted incorrectly had genomic HRD scores close to the clinical cut-off of 42, for SNP array data. MultiscaleHRD and BRCAness made incorrect predictions in samples with a wider range of HRD scores, while this was not the case for IdentifiHR (Fig. 4A). ExpHRD predicted most samples as HRD, despite having a predicted HRD score that was correlated ($R = 0.69$) with the true genomic HRD score (Supplementary Fig. 12A). This was a consequence of thresholding their predicted scores at the nominal value of 42 and suggested that with an adjusted HRD score cut-off, the accuracy of the expHRD model could be improved. Therefore, we optimised the threshold used to classify expHRD's predicted HRD score into a discrete HR status, taking the threshold value that would maximise the model's accuracy. With the optimal threshold of 84, expHRD did not outperform IdentifiHR, with the model correctly predicting the HR status of 54 samples, compared to IdentifiHR's 62, in the TCGA testing cohort ($n = 73$) (Supplementary Fig. 12B).

As the TCGA HGSC cohort was used to train expHRD (among other cancers) and the model's authors have not published which samples were used in training, there is likely overlap between their training cohort and our testing cohort. Given this has the potential to favourably bias expHRD in benchmarking, its performance in the AOCS testing cohort was also assessed. IdentifiHR had superior accuracy to expHRD, being 86% and 50% respectively, in the AOCS testing cohort, with 90 of the 99 samples being predicted to be HRD by expHRD using the clinical threshold of 42. Both models correctly predicted the HR status of all samples collected after autopsy, though while IdentifiHR also correctly predicted 8 of the 12 ascites-derived samples, expHRD predicted all to be HRD, meaning 6 were labelled incorrectly. All 7 samples from normal fallopian tube tissue were correctly predicted by both models; however, we noted that two samples were assigned negative HRD scores of −14.91 and −11.90 by expHRD, which is impossible for genomic HRD scores and complicates interpretation. Further, two other fallopian tube samples had HRD scores predicted by expHRD close to the clinical cut-off of 42, being 33.21 and 31.01, suggesting that despite these samples being taken from normal tissue, the model predicted that they had substantial genomic damage. In contrast, the probability of being HRD assigned by IdentifiHR in all fallopian tube samples of AOCS was less than 0.01. These results highlight that the scores predicted by the model do not perfectly represent genomic HRD scores derived from SNP arrays. However, the predicted HRD score of expHRD was moderately correlated with the genomic HRD score, from HRDsum ($R = 0.54$, $p$ value = $3.5 \times 10^{-8}$). After applying the optimal thresholding of HR status at 84, derived from the TCGA testing data, expHRD correctly predicted the HR status of 61 of the 99 AOCS cohort samples, though the model's accuracy, at 68%, was still below IdentifiHR's accuracy.

## Discussion

HRD remains the only molecular feature of HGSC that is clinically targetable. Consequently, the detection of HRD in HGSC patients, who are likely to benefit from PARPi treatment, remains of critical importance. While genome-based methods have had success in determining the historic status of HR in HGSC and represent the clinical gold standard, currently, there are no gene expression-based tools dedicated specifically for HGSC that have been built to predict HR status. We have built IdentifiHR to accurately predict HR status in HGSC using only gene expression. IdentifiHR can predict the HR status of a single RNA sequencing sample or an entire cohort. It assigns these predictions using a signal in gene expression that preserves

genomic information, such as HR status-specific CNVs and outputs both a discrete HR status, being HRD or HRP, in addition to the probability of the sample belonging to the HRD group. We show that the predicted probabilities correlate with the HRD score, which is continuous in nature. Beyond the HRD score, we show strong concordance between IdentifiHR's HR status predictions and the CHORD model, which utilises WGS and provides an alternate genomic source of the true HR status. A high degree of concordance was also observed between CHORD and the matched HRD score. As the genomic scarring that results from HRD has been shown to effectively classify HGSCs, it is important that a transcriptomic signature is equally robust in capturing a reliable signal. Our DE analysis of HRD compared to HRP has demonstrated that gene expression can reliably capture recurrent copy number changes associated with HR status, and based on the 209 genes with non-zero weights in our model, can be used to predict HR status in HGSC. We observe genomic regions with purely up- or downregulation of gene expression. While we have validated that some of these regions have HR-specific copy number changes, such as 8q24.2, 5q13.2 and 19q12, other regions of differential gene expression have undocumented aetiologies and motivate further investigation. The understanding of HRD fragile regions and the genomic and transcriptomic scarring that results from HRD remains incomplete across all cancers, though it should be the motivation for further study.

We provide, to our knowledge, the only direct comparison of transcriptomic HR status prediction tools in HGSC, against HR status, as determined by the clinical gold standard, being an HRD score. Benchmarking against other gene expression-based signatures to predict HR status, being BRCAness and MultiscaleHRD, in addition to the expHRD model, indicated that IdentifiHR was more accurate in predicting HR status in HGSC. This is in part likely to be because both BRCAness and MultiscaleHRD signatures were optimised to predict HR status in breast cancer, and while expHRD was built for a pan-cancer setting, it, like the other methods, is not HGSC specific. IdentifiHR is HGSC-dedicated tool, which is a non-trivial distinction, especially when considered alongside the incidence of HRD, as it most commonly occurs in HGSC compared to other cancers. Accordingly, IdentifiHR can capture the unique genomic and transcriptomic features of the disease and has not been tested in alternative cancers. BRCAness is limited by being a gene set, not a classification method, meaning the optimal way to infer HR status from it is unclear, and unlike IdentifiHR, which can accurately predict the HR status of non-*BRCA1/2* mediated HRD, the BRCAness signature was developed for *BRCA1/2* mediated HRD. However, we identified two cases within the TCGA testing cohort that had germline or somatic mutations in *BRCA1/2* yet had HRD scores less than the threshold of 42, and notably, the BRCAness signature, unlike IdentifiHR and MultiscaleHRD, predicted these cases to be HRD. While the true functional HR status of these retrospective samples cannot be determined, this does support the sensitivity of the BRCAness signature to alterations in *BRCA* genes, though it did also suggest that two cases with mutations in *BRCA1/2* and HRD scores over 42 were HRP. Alike IdentifiHR, expHRD was published as a predictive tool; however, we demonstrate that the HRD scores predicted by the model do not truly represent genomic HRD scores, and without optimisation of the threshold to define a discrete HR status, the model is largely inaccurate. Where possible, a hierarchical approach could be applied to optimise HR status determination, whereby founder mutations in known HR genes, such as *BRCA1/2*, could be independently assessed prior to using IdentifiHR. However, as we demonstrate, IdentifiHR remains largely robust regardless of mutation status, reinforcing its utility as a reliable tool across varied genetic contexts. CHORD and HRDscore remain the gold standard for HR status determination when genomic sequencing data is available. Where gene expression data is also available, IdentifiHR may be used to complement predictions.

While gene expression is not routinely assessed in the clinical management or treatment of HGSC, profiling gene-expression has shown success in the clinical management of many cancers and gene expression-based classifiers are of great research significance[51–55]. Our model demonstrates that the prediction of HR status in single cell sequencing data of HGSC is

possible after pseudo-bulking a small number of cells, though this may result in a loss of heterogeneity and composition bias. Nonetheless, it demonstrates that the application to spatial RNA sequencing may therefore be possible. This is an exciting prospect as it would allow for the investigation of HR at the spatial subclonal level, which may reveal reversion events and further our understanding of PARPi response. Investigation into subclonal populations of HGSC that have had a restoration of their HR mechanism is of vital importance, as these cells are highly unlikely to respond to PARPi treatment and will likely persist as treatment-resistant populations. As PARPi response is of clinical significance, predicting response as opposed to HR status should be explored. Currently, predictive models do not accurately predict functional HRD in any cancer. Our gene expression classifier uses features related to genomic scaring, and its performance on reversion events of HR is unknown. This limitation comes from a lack of reversion data. As RNA sequencing of patient cohorts at relapse increases and more training data becomes available, we anticipate that future gene expression classifiers will be able to identify both the historic and current HRD status of HGSC, and therefore predict PARPi response.

## Conclusions

IdentifiHR is an elastic net logistic regression model to predict HR status in HGSC, specifically, based on 209 genes. It captures both genomic signal and novel expression patterns, and outperforms existing gene expression models that can be used to predict HR status in HGSC. IdentifiHR provides foundational evidence to support further investigation into HRD fragile regions of the genome and the pan-cancer classification of HRD. We have built IdentifiHR as an R package to support use among the research and clinical community.

## Data availability

The results published here are in whole or part based upon data generated by The Cancer Genome Atlas, managed by the NCI and NHGRI. Information about TCGA can be found at http://cancergenome.nih.gov. RNA sequencing, gene-level copy number, methylation, SNP and structural variant data collected on the TCGA HGSC cohort, with associated clinical data, are available from the Genomic Data Commons (TCGA project) data portal (https://portal.gdc.cancer.gov/, https://www.cancer.gov/tcga, dbGaP Study Accession: phs000178.v11.p8). AOCS gene expression counts were accessed at the Gene Expression Omnibus (accession: GSE209964). Previously published WGS data are available from the European Genome-phenome Archive (accession: EGAD00001000877). MSKCC gene expression counts were available as a Seurat object on Synapse (SynID: syn51091849). The source data for Fig. 2A, D, E can be found in Supplementary Data 2, for Fig. 3A, B in Supplementary Data 5, for Fig. 3C in Supplementary Data 6, for Fig. 3D, E in supplementary data 7, for Fig. 3F in Supplementary Data 9 and for Fig. 4A–C in Supplementary Data 5. These data are available in the supplementary information and in the IdentifiHR repository, https://github.com/DavidsonGroup/IdentifiHR. All other data supporting the findings of this study, including the source data for all figures, are publicly available.

## Code availability

All analyses were carried out in R v4.2.1. Code to reproduce the analysis can be found in the IdentifiHR repository, https://github.com/DavidsonGroup/IdentifiHR.

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

## Acknowledgements

A.L.W. is supported by a Research Training Program scholarship and is partially funded by a CSL PhD top-up scholarship and a Tour De Cure PhD grant. N.M.D. is funded by NHMRC Investigator Grant [GNT2016547 to N.M.D.] and the Estate of Judith Corrie Philpots. S.J.R. is funded by NHMRC Investigator Grant [GNT2009840 to S.J.R]. We thank Dr Matthew Wakefield for offering insight and expertise in ovarian carcinoma biology. We acknowledge the contributions of Dr Ksenija Nesic and the entire laboratory of Professor Clare Scott at the Walter and Eliza Hall Institute for offering feedback on the complete IdentifiHR model. We offer thanks to Professor James Brenton and members of the Brenton laboratory for discussions surrounding HR and the training of our model. Figures 1 and 3 created, in part, in BioRender. Weir, A. (2025); https://BioRender.com/isms2aw. We thank the many patients who contributed to the data used in this research, and our cancer consumer advisers. We also acknowledge the Wurundjeri people of the Kulin nation as the traditional owners and guardians of the land on which the work was performed.

## Author contributions

A.L.W. and N.M.D. conceived and designed the study. A.L.W. collected, processed, and curated all data, developed the methodology, validated the method and results, wrote the original draft and all subsequent iterations, and produced all tables and visualisations in the study. N.M.D., S.J.R. and C.W.T. supervised the research and revised the manuscript. D.G and A.P. processed and analysed WGS data in the AOCS cohort. S.C.L. and M.L. advised on method development and analysis. All authors contributed to the review of the manuscript. All authors approved the manuscript for submission.

## Competing interests

The authors declare no competing interests.
