## [Transparent Peer Review file · Communications Medicine]

IdentifiHR predicts homologous recombination deficiency in high-grade serous ovarian carcinoma using gene expression

Corresponding Author: Dr Nadia Davidson

Version 0:

Reviewer comments:

Reviewer #1

(Remarks to the Author)

The authors created a model to infer HRD status calls from RNAseq data in a HGSC cohort. The model was build based on TCGA data and verified with an independent cohorts as well as pseudobulked single-cell data. The high accuracy in the AOCs cohort shows good generalisability.

The study is limited by the definition of HR deficiency, preferably the cases could be separated by known deleterious BRCA mutations + hypermethylated cases vs. known wildtype cases. Maybe there are not enough samples for that in the cohort. The authors used scarHRD for the calling of HRD scores. Since the definition of HRD in the training cohort has a large influence the usage of methylation data and referencing a second method such as the SNP array data could be used to strengthen the definition of true HRD for training (for an example of more detailed case separation see Rempel et. al 2022 NPJ Prec. Oncol.)

From the methods it is unclear how the TCGA HRD scores were calculated, scarHRD is mentioned, however, this tool requires allele-specific copy numbers as input, often calculated using Sequenza. Which tool was used for this step?

Presumably a connection between (high-level) amplifications and increased expression is observable, since the amplified regions are larger than single genes, are genes in close vicinity of each other ranked similarly in the HRD prediction model?

Tumor purity was tested as a confounder in the AOCs cohort, could a limitation of identifyHR be observed in the TCGA cohort? E.g. were samples with lower tumor purity more difficult to asses?

Minor:

Figure 4 (and others) color scales are empty

Reviewer #2

(Remarks to the Author)

The authors present a predictor of homologous recombination deficiency (HRD) based on gene expression for high-grade serous ovarian carcinomas (HGSCs). The method named IdentifiHR is based on an elastic net penalized logistic regression model which was trained on 288 HGSC samples from TCGA (160 HRD, 128 non-HRD), and evaluated based on three different test data sets (TCGA: n=73; AOCs: n=99; MSKCC: n=37).

Further, the performance of IdentifiHR was compared to existing HRD classifiers based on gene expression data, namely using differential expression in BRCAness-related genes as proposed by Guo and Wang, MultiscaleHRD, and expHRD. IdentifiHR outperformed these approaches with respect to accuracy, precision, and misclassification error.

The manuscript is very well-written, and the methods and results are thoroughly and comprehensively presented.

It is regrettable that no comparison of the performance of IdentifiHR with the classic methods of HRD testing based on

variant data is possible, but I realize that this would go beyond the scope of this study. However, it should be shortly mentioned, if and how co-expression affects the model, e.g., was co-expression of genes considered and could BRCA1/2 not be included in the final model as there are highly correlated genes serving as covariates?

Beyond that, I can only point to a single typo (Keywords: Gene expression -> gene expression), and have no further comments.

Reviewer #3

(Remarks to the Author)

In this study, the authors developed an innovative, dedicated HRD detection tool (IdentifiHR) based on applying a machine learning classifier to gene expression data. I would like to congratulate the authors for finding such a valuable research gap, as well as for their detailed background description and robust development of their application. In addition, I can confirm that I was able to successfully run their tool, which was also nicely documented and packaged in an R environment. I have now provided some comments and suggestions, as well as requested further clarifications and evaluations to determine novelty.

In this regard, my main concern is that the authors should refrain from mentioning that “there are currently no methods to classify HR status in HGSC using gene expression” since, as they acknowledged (and benchmarked), at least two pan-cancer methods have been developed previously (based on elastic net penalized regression and lasso logistic regression). Instead, the authors should emphasize the lack of “dedicated” methods for HGSC, and what the benefits—and also caveats—of such dedicated methods are, as well as warn users about the HGSC-dedicated nature of their algorithm.

The rest of my comments and suggestions can be found below.

Major comments

1. Regarding the differential expression analysis, it is unclear why the authors did not include a threshold for the fold change of the differentially expressed genes, instead of directly using all ~2,600 genes. I suspect there might be irrelevant genes in the HRD-like signatures derived from these, especially those having low fold change values. The authors should evaluate whether removing some of these genes substantially changes their HRD detection method.
2. In addition, the relevance of some of the results provided is also unclear. For example, the importance of the genomic distribution of differentially expressed genes shown in Fig. 2D, Fig. 2E, or Supplementary Fig. 3C is not sufficiently discussed. Similarly, the authors discussed the presence of frequently altered genes in the same genomic regions as differentially expressed genes, but have not checked if these are actually altered in the same samples. The authors should carefully evaluate which are the relevant results of their analysis for these to be highlighted in the main figures, as well as provide more compelling evidence for the hypothesis of co-occurrence of genomic and transcriptomic cancer driver events in the same genomic regions.
3. The authors should clarify what the main novelty of their newly developed method (besides its application specifically to HGSC) is, compared to the previous pan-cancer methods for HRD detection based on gene expression. Indeed, IdentifiHR seems to be a simpler approach (predicting general HRD) compared to more sophisticated methods, like MultiscaleHRD, which can predict a BRCA1 or BRCA2-like HRD phenotype. Also, they should do a better job in describing the selection of their optimal hyperparameters, which at the moment seems a bit arbitrary in the case of the alpha value fixed at 0.53 without any explanation (as it does not seem to lead to the highest AUC value according to the figure—0.28 seems to be the highest). They should also clarify why they decided to fix the threshold for HRD/HRP of their method at 50%, instead of only classifying as HRD those above a certain threshold and HRP below another lower threshold, with values in the middle considered as uncertain.
4. IdentifiHR should not only be compared to gene expression-based classifiers, but also to genomic-based ones, such as the mentioned CHORD or HRDetect, as well as other more recent ones that the authors have missed and should also include, like SigMA (<https://doi.org/10.1038/s41588-019-0390-2>) or HRProfiler (<https://doi.org/10.1101/2024.07.14.24310383>). For this, the authors should explore alternative ground truth definitions for HRD, including mutations in BRCA1/2, mutations in all HR pathway genes (including others like RAD51, PALB2, etc.), HRD scores based on copy number, and HRD scores based on mutational signatures. This multi-ground truth analysis and comparison against other classifiers will deepen their manuscript and provide more robust evidence of the usefulness of their approach in comparison to others currently available in the literature, independently of the input data used. Ideally, the authors should also test how their method predicts platinum and/or PARP inhibitors sensitivity compared to other HRD scores or mutations. Considering that probably none of the approaches is perfectly capturing all cases susceptible to benefit from these therapies, a direct assessment would be the ideal scenario to test their clinical applicability.
5. The different ground truths used for determining HRD status between the discovery and validation cohorts make comparisons difficult. The authors should make sure to evaluate the discovery and validation cohorts using at least one similar ground truth, such as the HRD score derived from copy number alterations or the CHORD score based on mutational signatures, even if the original authors of the publications where they derived the data used a different one.

6. The finding observed in the AOCs cohort that some of the cases predicted to be HRP by IdentifiHR but expected to be HRD harbored HRD-related somatic mutations, including mutations in BRCA1 and BRIP1, suggests that maybe the combination of data from mutations and gene expression could achieve better results than gene expression alone. Have the authors considered the possibility of including mutation information in their method for those cases where this is available? Presumably, many HGSC cases will have some sequencing data available in a clinical setting, at least from panel sequencing (e.g., through MSK-IMPACT).

7. Although, as previously mentioned, the research gap addressed seems interesting and of great potential, using just TCGA as the discovery dataset seems a bit outdated, as different multi-omics databases have now been constructed. It would be ideal if the authors could explore other large databases for this study, at least as validation cohorts. Examples of large studies are Hartwig (based on Dutch patients; ~7,000 metastatic cancers), PCAWG (mostly based on US and European patients; ~2,800 primary tumors), and others.

Minor comments

1. Despite the very detailed background section provided, which is indicative of a nice research of the state of the art, I suggest the authors to reduce this introductory section to benefit readability. In particular, they should refrain from extensive descriptions of previously published results (such as in their fourth paragraph), leaving readers to explore associated publications instead.

2. On a related note, regarding copy number signatures, the alternative framework to the one developed by Drews et al., provided by Steele et al. (<https://doi.org/10.1038/s41586-022-04738-6>), should also be indicated.

3. Similarly, other recent machine learning classifiers for HRD based on genomics data are missing, including SigMA (<https://doi.org/10.1038/s41588-019-0390-2>) and HRProfiler (<https://doi.org/10.1101/2024.07.14.24310383>).

4. References should also be added to the abstract, as well as in some statements of the results referring to prior literature (e.g., line 413).

5. In the Methods section, the authors should clarify the source of the TCGA metadata, as well as if BRCA1/2 mutations were single or double hits.

6. The pseudobulk processing of the scRNA-seq expression data from the MSKCC, as well as the potential caveats of the use of pseudobulk single-cell data, should be further discussed in the Methods and Discussion sections.

7. Methods applied for the adjustment of p-values across the manuscript should be explicitly mentioned and cited.

8. The use of two levels of headings and subheadings within the Results section is unclear and too complex. Please use just a single level of headings within the Results.

9. The diagram in Fig. 1 is missing an indication that the final number of genes considered after the elastic net penalization is 209.

10. It is currently impossible to understand the difference between Fig. 2B and Fig. 2C without looking at the legend. That is not a good practice, and the authors should consider adding titles and/or subtitles to their plots to avoid this.

11. Also, Fig. 2C is mentioned in the text before Fig. 2B, which is also not a good practice. Please reorder the figures accordingly if needed.

12. Furthermore, some quantitative metric should be used for comparing PCAs in Fig. 2B and 2C beyond the current mention of visual separation.

13. The order of panels in Fig. 3 is confusing. Why is panel D at the bottom?

14. The legend for Fig. 3F is incomplete (the colors for IdentifiHR probability are missing, at least in Mac Preview). Also, these colors seem very similar to the colors used for the Patient HR status. Please update this to avoid issues with the interpretation of the figure.

15. Similarly, the legend for Fig. 4A is incomplete (missing colors for HRD score).

16. Supplementary table legends are missing, at least in the version shared for peer review.

17. The mention of supplementary table 6 ("IdentifiHR is an elastic net penalized logistic regression model based on the expression of 209 protein coding genes" section of the Results) is confusing. Shouldn't this reflect the predictions in the TCGA cohort? Instead, there is an indication of the 209 genes, which should be referenced later in the paragraph.

18. The installation of the package (at least on a Mac system) generated an error due to the suggested `build_vignettes = TRUE` option. Setting this flag to `FALSE` seems to solve this issue. In addition, it seems that the `SummarizedExperiment`

package is also needed and was not included in the default installation option. Although this could be solved by most experienced users, a smoother installation is needed and, preferably, as part of Bioconductor.

Version 1:

Reviewer comments:

Reviewer #1

(Remarks to the Author)

The author addressed all of my comments satisfactory.

Reviewer #2

(Remarks to the Author)

The authors have satisfactorily incorporated my comments into the revised version of the manuscript. I have no further concerns.

Reviewer #3

(Remarks to the Author)

I would like to sincerely thank the authors for their comprehensive revision of the manuscript, incorporating the feedback provided by reviewers. Although most of my comments have been successfully addressed, I still have a few additional minor comments regarding my prior requests (I will be referring to them using the previously provided numbering for major/minor comments):

Major comments

3. The authors have now outlined several advantages of IdentifiHR for its usage in HGSC specifically. Considering they have not specifically indicated in their response the changes made to the main manuscript with respect to each of the comments, they should ensure to include these additional details of IdentifiHR advantages in both the Results and the Discussion sections. I strongly recommend that they include the parts of the manuscript changed in a third color in their rebuttal documents for future publications to help peer reviewers.

Regarding detection thresholds, I appreciate their effort to derive a cohort-specific threshold (although they have not provided the details about how this was done in their response). However, they still did not try my original suggestion to derive two independent thresholds for detection of HRD and HRP (e.g., 75% and 25%), with values in the middle determined as uncertain. I still think this could lead to more clinically meaningful results, although this is a minor point for the overall manuscript. The authors should at least indicate why this is or not a suitable suggestion in their opinion.

4-5. Although understanding the authors' rationale of a gene expression classifier, according to their results CHORD seems to have a better performance than their IdentifiHR approach. Although they focused on the consistency/concordance between methods (Supplementary Figure 8), it's important to acknowledge that CHORD shows better results at least in the AOCs testing cohort (accuracy 88%, 65/74, vs. 85%, 63/74; according to Supplementary Figure 9E), which is the only place where it was possible to directly review CHORD's performance. The authors should also include CHORD's prediction results in the TCGA-OV testing cohort panel in Fig. 4A (as they did in Supplementary Figure 9E), as well as a similar table as the one provided for their IdentifiHR tool in Fig. 3B. The current table in Supplementary Figure 8B compares CHORD vs. IdentifiHR, but not CHORD vs. their ground truth (HRD score derived from SNP array data with a threshold of 42). Also, it is unclear why only 43 samples were included in Supplementary Figure 8B vs. 73 in Figure 3B for the TCGA testing cohort, as all cases in TCGA should have sufficient genomic data for evaluating HR status through CHORD.

Considering the results of CHORD compared to IdentifiHR in the AOCs cohort, as well as the additional evidence requested for TCGA, the authors should include in their discussion a mention that despite the potential usefulness of a gene expression exclusive classifier, current genomic based HR classifiers like CHORD seem to have slightly superior or, at least, equivalent performance.

Minor comments

14. Thanks for revising the color scale for the IdentifiHR color scale. However, the color palette has not been updated to avoid confusion with the patients' HR status and the HRD score in other figures. Considering the different greens used for HRP and HRD status according to IdentifiHR, probably a scale of greens would be better to avoid confusion and be consistent with other figures.

15. Now the color scale for the BRCA1 minor allele CN is incorrect (currently displayed as the same color for all copy number values).

Version 2:

Reviewer comments:

Reviewer #3

(Remarks to the Author)

I thank the authors for satisfactorily incorporating my comments into the revised version of the manuscript. I have no further concerns.

IdentifiHR: predicting homologous recombination deficiency in high-grade serous ovarian carcinoma using gene expression.

We thank the reviewers for their constructive feedback and appreciate the opportunity to improve our manuscript. We have now revised our manuscript in accordance with the suggested changes. Major changes to the manuscript include:

- A more comprehensive exploration of alternative ground truth definitions for HRD, specifically considering *BRCA1* and *RAD51C* promoter hypermethylation, minor allele *BRCA1* copy number (reflecting a loss of heterozygosity) and CHORD HR status predictions.
- An evaluation of the relationship between gene expression and copy number changes in HRD fragile regions of the genome (8q24.2, 5q13.2 and 19q12).
- An analysis of weighted model gene co-expression with *BRCA1/2*.
- Probability threshold optimisation for IdentifiHR, to define the probability at which the model's accuracy in predicting discrete HR statuses is highest.
- A direct comparison between TCGA and AOCs testing cohorts, using a HRD score and CHORD HR status predictions.

We have amended instances in the text where we had stated “there are currently no methods to classify HR status in HGSC using gene expression”, and instead emphasise that the novelty of our tool, IdentifiHR, is derived from it being a HGSC-dedicated method, and to highlight that this specificity results in more accurate predictions of HR status. During our revision, we found an error in our HR status labelling of the AOCs cohort which resulted in a minor decrease in IdentifiHR's accuracy from 91% to 86%. We have amended the relevant text, figures and tables. We have also included additional information about genomic methods of HR status prediction in the introduction and information in the methods detailing new analyses. Please find a detailed response to each of the reviewer's comments below.

Reviewers' comments:

Reviewer #1:

The study is limited by the definition of HR deficiency, preferably the cases could be separated by known deleterious *BRCA* mutations + hypermethylated cases vs. known wildtype cases. Maybe there are not enough samples for that in the cohort. The authors used scarHRD for the calling of HRD scores. Since the definition of HRD in the training cohort has a large influence the usage of methylation data and referencing a second method such as the SNP array data could be used to strengthen the definition of true HRD for training (for an example of more detailed case separation see Rempel et. al 2022 NPJ Prec. Oncol.)

We agree that our study would be enhanced by a more thorough examination of the epigenetic and genomic causes of HRD. Accordingly, we have assessed *BRCA1* and *RAD51C* promoter hypermethylation and confirmed that all cases with potential *BRCA1* or *RAD51C* silencing were labelled as HRD by the ScarHRD score across both the TCGA training and testing cohort (figure 4A, supplementary figure 1A-B). We have also examined germline and somatic mutations of *BRCA1/2* in our testing cohort. IdentifiHR predicted 11 of the 13 cases with *BRCA1/2* variants to be HRD. Neither of the 2 cases labelled as HRP with variants had founder, pathogenic mutations. One was a benign nonsense germline mutation in *BRCA2* (p.K3326*) and the other was a somatic *BRCA1* missense mutation (p.C47W) of uncertain functional significance, accompanied by a loss of heterozygosity, raising the possibility of biallelic inactivation. However, the low HRD score of 27 suggested the variant was unlikely to be pathogenic and that the sample was relatively HR proficient. We have included this information in the results of our manuscript (line 545-557) and proposed a hierarchical approach in our discussion for optimal HR status determination, whereby mutations could be independently assessed prior to using IdentifiHR. However, as we demonstrate, IdentifiHR remains robust regardless of mutation status, reinforcing its utility as a reliable tool across varied genomic contexts (line 631-635).

From the methods it is unclear how the TCGA HRD scores were calculated, scarHRD is mentioned, however, this tool requires allele-specific copy numbers as input, often calculated using Sequenza. Which tool was used for this step?

We appreciate the reviewer bringing this oversight to our attention, estimates of ploidy and absolute allelic copy-number used to derive the HRD score were estimated using ABSOLUTE (Carter, S.L. et al., 2012, Nature Biotechnology). We have now included this information in the “Data sources: The TCGA cohort” section of our methods (line 191-192).

Presumably a connection between (high-level) amplifications and increased expression is observable, since the amplified regions are larger than single genes, are genes in close vicinity of each other ranked similarly in the HRD prediction model?

We have now examined the connection between high-level amplification and increased expression, using three representative genomic regions known to have HR status-dependent associations (8q24.2, 5q13.2 and 19q12). We examined this relationship in genes determined to be significantly differentially expressed in our analysis. A significant, though moderate, correlation was observed between absolute gene-level copy number and matched gene expression, in HRD and HRP cases, at 8q24.2 and 5q13.2, whereby expression increased with copy number. This was also observed in HRP cases, though it was less pronounced in HRD cases, at 19q12. Supplementary figure 4A-C presents these findings and supplementary figure 6 highlights the role of this biology in our model, and they are briefly discussed in the manuscript (line 373-378).

Further, we can clarify that while the mean copy number and expression fold change of genes in close genomic proximity is somewhat similar, by HR status (supplementary figure 4D, 6A-F), the model weights of genes used by IdentifiHR vary, regardless of being in close vicinity to each other, as illustrated by supplementary figure 5D and supplementary figure 6G-H. This is expected, as while some of the model features are representatively of genomic scarring, they should still be relatively independent.

Tumor purity was tested as a confounder in the AOCS cohort, could a limitation of identifyHR be observed in the TCGA cohort? E.g. were samples with lower tumor purity more difficult to assess?

As suggested, we have assessed the tumour purity of TCGA HGSC samples as a potential limitation of IdentifiHR. TCGA HGSC samples with correctly predicted and incorrectly predicted HR status labels, by IdentifiHR, both had high mean purity levels (86.2% and 84.7% respectively) and did not significantly differ (Welch’s two-sided t-test, $t = 0.47582$, $df = 13.225$, $p\text{-value} = 0.642$). This is consistent with what we observed in the AOCS testing cohort. We have updated the results of manuscript to include this analysis (line 424-427).

Minor:

Figure 4 (and others) color scales are empty

We apologise for this formatting issue as our manuscript was submitted with completed colour scales. We have regenerated this figure and checked the revised manuscript in the hope of resolving this issue.

Reviewer #2 (Remarks to the Author):

It is regrettable that no comparison of the performance of IdentifiHR with the classic methods of HRD testing based on variant data is possible, but I realize that this would go beyond the scope of this study. However, it should be shortly mentioned, if and how co-expression affects the model, e.g., was co-expression of genes considered and could *BRCA1/2* not be included in the final model as there are highly correlated genes serving as covariates?

As suggested, we have now explored potential co-expression between the genes in our final model and *BRCA1/2*. We have included these findings in our results section (line 401-410), and note that genes were weakly to moderately correlated with *BRCA1* and 2, though no individual model genes were strongly correlated with the expression of either gene (supplementary table 6). A significant, linear association was however observed between the beta coefficient for each model gene and its matched correlation coefficient with *BRCA1*, but not *BRCA2*, expression (supplementary figure 5E-F). This is notable as it suggests that while an individual gene may not be strongly co-expressed with *BRCA1/2*, the model does capture some signature of *BRCA1* expression.

Beyond that, I can only point to a single typo (Keywords: Gene expression -> gene expression), and have no further comments.

This mistake has now been corrected (line 42).

Reviewer #3 (Remarks to the Author):

In this regard, my main concern is that the authors should refrain from mentioning that “there are currently no methods to classify HR status in HGSC using gene expression” since, as they acknowledged (and benchmarked), at least two pan-cancer methods have been developed previously (based on elastic net penalized regression and lasso logistic regression). Instead, the authors should emphasize the lack of “dedicated” methods for HGSC, and what the benefits—and also caveats—of such dedicated methods are, as well as warn users about the HGSC-dedicated nature of their algorithm.

We have amended all instances where we have failed to clarify that IdentifiHR is a tool developed specifically for HGSC, and the use of which is dedicated to only HGSC (line 120-121, 135-137, 595-596, 618-620, 654).

The rest of my comments and suggestions can be found below.

Major comments

1. Regarding the differential expression analysis, it is unclear why the authors did not include a threshold for the fold change of the differentially expressed genes, instead of directly using all ~2,600 genes. I suspect there might be irrelevant genes in the HRD-like signatures derived from these, especially those having low fold change values. The authors should evaluate whether removing some of these genes substantially changes their HRD detection method.

We did not apply a fold-change threshold to reduce our feature space from the 2604 genes identified in differential expression analysis, as we wanted to ensure an unbiased selection of genes in our final model. A smaller fold-change between HRD and HRP cases does not necessarily reflect a small or irrelevant power to separate HRD and HRP. Instead, we relied on statistical significance, which is a combination of fold-change and evidence (expression). We have instead utilised elastic net penalisation of the 2604 input genes used in training to ensure that only those that meaningfully contribute to distinguishing HRD from HRP in HGSC were included in our final model, irrespective of fold-change.

2. In addition, the relevance of some of the results provided is also unclear. For example, the importance of the genomic distribution of differentially expressed genes shown in Fig. 2D, Fig. 2E, or Supplementary Fig. 3C is not sufficiently discussed. Similarly, the authors discussed the presence of frequently altered genes in the same genomic regions as differentially expressed genes, but have not checked if these are actually altered in the same samples. The authors should carefully evaluate which are the relevant results of their analysis for these to be highlighted in the main figures, as well as provide more compelling evidence for the hypothesis of co-occurrence of genomic and transcriptomic cancer driver events in the same genomic regions.

Some similar comments were raised by reviewer #1 (third comment), please also see the response given there.

To emphasise the importance of the genomic distribution of differentially expressed genes, we have included analysis of the relationship between absolute gene-level copy number and expression in the chromosome regions 8q24, 5q13 and 19q12, which have previously been shown to harbour HR status specific copy number variations (supplementary figure 4 and 6). We demonstrate that copy number and expression are significantly, positively correlated in these regions in the same samples, and when the mean of each is taken across the TCGA training cohort, can be used to clearly distinguish HRD cases from those that are HRP (though this was less pronounced at 19q12) (supplementary figure 4A-D). We confirmed our assumption of co-occurring genomic and transcriptomic cancer driver events with these findings and have clarified the text in our results (line 373-378). Further, we have removed the text mentioning specific genes, “which contains the oncogenes *MYC* and *NDRG1*”, in reference to the 8q24.2 genomic region to avoid confusion.

3. The authors should clarify what the main novelty of their newly developed method (besides its application specifically to HGSC) is, compared to the previous pan-cancer methods for HRD detection based on gene expression. Indeed, IdentifiHR seems to be a simpler approach (predicting general HRD) compared to more sophisticated methods, like MultiscaleHRD, which can predict a BRCA1 or BRCA2-like HRD phenotype. Our comparative analysis of gene expression methods to predict HR status in the TCGA testing cohort highlights the novelty of IdentifiHR in predicting HR status in HGSC specifically. As none of the other methods had been trained to account for the unique biology or extreme heterogeneity of HGSC, IdentifiHR not only outperformed all other methods but was the only method that appeared to be appropriate for use in these tumours (please see results section “*IdentifiHR outperforms BRCAness, MultiscaleHRD and expHRD*”). IdentifiHR’s HGSC specificity is a non-trivial distinction, especially when considered alongside the incidence of HRD, as it most commonly occurs in HGSC compared to other cancers. In addition, while a webservice interface has been developed to support the use of expHRD, BRCAness and MultiscaleHRD have no developed tools to support use. IdentifiHR has been developed as a user friend R package to be submitted to Bioconductor.

Also, they should do a better job in describing the selection of their optimal hyperparameters, which at the moment seems a bit arbitrary in the case of the alpha value fixed at 0.53 without any explanation (as it does not seem to lead to the highest AUC value according to the figure—0.28 seems to be the highest).

We appreciate that our manuscript could be enhanced by clarifying the selection of optimal hyperparameters, and while we had mentioned this in our methods section (line 280-282), have also added a brief sentence to our results to ensure clarity (line 387-389).

They should also clarify why they decided to fix the threshold for HRD/HRP of their method at 50%, instead of only classifying as HRD those above a certain threshold and HRP below another lower threshold, with values in the middle considered as uncertain.

Our decision to set the probability threshold for a sample being HRD at 0.5 in our model stemmed from this yielding approximately the expected HGSC proportion of each HR class, being ~50%, in each of our use cases. As our tool outputs the probability that corresponds to each HR status prediction, users can adjust this threshold based on their own expectations in unique datasets. However, we agree that the optimal threshold should be considered and accordingly, utilised the TCGA testing cohort for optimisation. The optimal threshold was found to be 0.58 and increased the accuracy in the TCGA testing cohort from 85% to 86% (supplementary figure 10A). However, applying this threshold to the AOCS testing cohort for independent assessment, reduced accuracy from 86% to 85% (supplementary figure 10B-C). In both cohorts, the change resulted in the re-classification of only a single sample. Given this discrepancy, the probability threshold of 0.5 is still used in our package. We have included this discussion in our results (line 479-486).

4. IdentifiHR should not only be compared to gene expression-based classifiers, but also to genomic-based ones, such as the mentioned CHORD or HRDetect, as well as other more recent ones that the authors have missed and should also include, like SigMA (<https://doi.org/10.1038/s41588-019-0390-2>) or HRProfiler (<https://doi.org/10.1101/2024.07.14.24310383>). For this, the authors should explore alternative ground truth definitions for HRD, including mutations in BRCA1/2, mutations in all HR pathway genes (including others like RAD51, PALB2, etc.), HRD scores based on copy number, and HRD scores based on mutational signatures. This multi-ground truth analysis and comparison against other classifiers will deepen their manuscript and provide more robust evidence of the usefulness of their approach in comparison to others currently available in the literature, independently of the input data used.

While we agree that exploring alternate ground truth definitions for HRD is valuable, the aim of our study was to develop a method that used only gene expression to predict HR status in only HGSC. We have accordingly benchmarked IdentifiHR against gene-expression based signatures and methods that can also be applied in this setting. We have considered the performance of IdentifiHR against the clinical gold standard for HR status determination, being a HRD score (derived from SNP array data with a threshold of 42) and HR gene mutations, with a focus on those involving BRCA1/2 (figure 4). In our revised manuscript, we have also considered BRCA1 and RAD51C promoter hypermethylation and BRCA1 heterozygous loss in copy number as additional genomic causes of HRD, and have compared IdentifiHR against CHORD, in both the TCGA and AOCS testing cohorts (figure 4, supplementary figure 8-9).

To extend our comparison to alternate genomic methods for HR status prediction is beyond the scope of this project, since the most recent of those listed by the reviewer, being HRProfiler, provides a comprehensive assessment of their relative strengths and weaknesses in their article, and suggests HRProfiler and CHORD to be generally superior to alternate methods. Both HRProfiler and IdentifiHR have a high concordance with CHORD predictions, likely since HRProfiler uses a reduced set of mutational features similar to that employed by CHORD. Further, HRProfiler's ovarian-specific model was trained on the TCGA ovarian cohort and used a HRD score (with a threshold at 63, not 42) and germline/somatic alteration in *BRCA1/2* to define "true" HRD cases. We have shown that IdentifiHR correctly predicts all cases with a HRD score > 63 and germline/somatic mutations in *BRCA1/2* as "HRD".

Ideally, the authors should also test how their method predicts platinum and/or PARP inhibitors sensitivity compared to other HRD scores or mutations. Considering that probably none of the approaches is perfectly capturing all cases susceptible to benefit from these therapies, a direct assessment would be the ideal scenario to test their clinical applicability.

We completely agree that a signature that accurately predicts patient sensitivity to chemotherapy or PARP inhibitors would be of great research and clinical significance. Such a method requires access to large, well-annotated patient cohorts with matched gene expression, genomic sequencing (to determine the "true" HR status), mutational data, and detailed treatment response information, especially in the context of PARP inhibitor treated HGSC patients. Unfortunately, treatment response was not comprehensively detailed in any of the cohorts available to us, though we concur that predicting treatment response would be ideal to extend the clinical applicability of future methods, and have commented on this in the discussion of our manuscript.

5. The different ground truths used for determining HRD status between the discovery and validation cohorts make comparisons difficult. The authors should make sure to evaluate the discovery and validation cohorts using at least one similar ground truth, such as the HRD score derived from copy number alterations or the CHORD score based on mutational signatures, even if the original authors of the publications where they derived the data used a different one.

We acknowledge the difficulty caused by using different ground truths for determining HR status and have now assessed both the HRD score and the CHORD HR predictions in both the TCGA and AOCs testing cohorts. In our revised manuscript, we have used CHORD to predict HR status in the subset of the TCGA HGSC cohort with WGS. We found the CHORD probability of a sample being HRD to be strongly, positively correlated with the matched, SNP-array derived HRD score ($R = 0.71$, $p\text{-value} = 2.2 \times 10^{-16}$) and found IdentifiHR predictions to be 79% concordant with CHORD (supplementary figure 8A-C). Further, we have collaborated with members of the AOCs group to access HRD scores for the AOCs testing cohort to allow for this direct comparison (supplementary figure 9). We found these scores to be slightly higher than expected (mean HRD score = 59.1) due to them being derived from whole genome sequencing rather than SNP arrays, and after optimising the threshold for discrete HR status prediction, found them to be 81% concordant with the matched CHORD HR status predictions (supplementary figure 9A-B). The accuracy of IdentifiHR against CHORD labels was 86% and against the genomic HRD score was 81% (supplementary figure 9C-D). We have incorporated comprehensive discussion of these findings in our results section (line 428-434, 442-460).

6. The finding observed in the AOCs cohort that some of the cases predicted to be HRP by IdentifiHR but expected to be HRD harbored HRD-related somatic mutations, including mutations in *BRCA1* and *BRIP1*, suggests that maybe the combination of data from mutations and gene expression could achieve better results than gene expression alone. Have the authors considered the possibility of including mutation information in their method for those cases where this is available? Presumably, many HGSC cases will have some sequencing data available in a clinical setting, at least from panel sequencing (e.g., through MSK-IMPACT). We were motivated to develop IdentifiHR by the lack of methods using only gene expression to predict HR status. We agree with the reviewer that a hierarchical approach considering the mutational status of known HR genes could extend the clinical applicability of our method and have mentioned this in our text (line 631-635). Though, as we have shown IdentifiHR to be robust regardless of mutation status, mutation information can be incorporated at the user's discretion. We wish to preserve the gene expression only approach of IdentifiHR to ensure it remains accessible to users who do not have mutation data available.

7. Although, as previously mentioned, the research gap addressed seems interesting and of great potential, using just TCGA as the discovery dataset seems a bit outdated, as different multi-omics databases have now been constructed. It would be ideal if the authors could explore other large databases for this study, at least as validation cohorts. Examples of large studies are Hartwig (based on Dutch patients; ~7,000 metastatic cancers), PCAWG (mostly based on US and European patients; ~2,800 primary tumors), and others. Ideally, we would have data from numerous large cohorts to train and test our model, however appropriate data for HGSC is limited. To the best of our knowledge PCAWG HGSC samples are included in the TCGA data we have used. We attempted to apply for the Hartwig data but have been unable to gain access due to legal advice related to being based in a non-collaborative country. We also note that Hartwig contained only 77 confirmed cases of HGSC, which is an analogous size to our two other independent testing cohorts, AOCs and MSKCC. Given this, we believe that our inclusion of two independent HGSC-specific validation cohorts already provides robust external validation, inline or even exceeding, the validation performed by other published models. Although older, we expect that the TCGA gene-expression and HRD scores are accurate.

Minor comments

1. Despite the very detailed background section provided, which is indicative of a nice research of the state of the art, I suggest the authors to reduce this introductory section to benefit readability. In particular, they should refrain from extensive descriptions of previously published results (such as in their fourth paragraph), leaving readers to explore associated publications instead.

We have reduced the size of the introduction, with a particular focus on the fourth paragraph.

2. On a related note, regarding copy number signatures, the alternative framework to the one developed by Drews et al., provided by Steele et al. (<https://doi.org/10.1038/s41586-022-04738-6>), should also be indicated.

We have now referenced the Steele et al. copy number framework in our introduction (line 98).

3. Similarly, other recent machine learning classifiers for HRD based on genomics data are missing, including SigMA (<https://doi.org/10.1038/s41588-019-0390-2>) and HRProfiler (<https://doi.org/10.1101/2024.07.14.24310383>).

We thank the reviewer for bringing HRProfiler to our attention, as it was not published prior to our initial submission. Both SigMA and HRProfiler are now mentioned in the introduction (line 106-107, 111-113).

4. References should also be added to the abstract, as well as in some statements of the results referring to prior literature (e.g., line 413).

The Communications Medicine journal guidelines preclude references in the abstract. We have however included additional references in the results section.

5. In the Methods section, the authors should clarify the source of the TCGA metadata, as well as if BRCA1/2 mutations were single or double hits.

We have now included this in our methods section (line 181-187).

6. The pseudobulk processing of the scRNA-seq expression data from the MSKCC, as well as the potential caveats of the use of pseudobulk single-cell data, should be further discussed in the Methods and Discussion sections.

Counts were pseudobulked by summing those for all cells of the same sample to the gene level. This information, and mention of the caveats of pseudobulking, has been added into our methods and discussion sections, respectively (line 222, 640).

7. Methods applied for the adjustment of p-values across the manuscript should be explicitly mentioned and cited.

We thank for reviewer for bringing this oversight to our attention and note that the method applied for the adjustment of p-values is now included in the manuscript (line 255-257).

8. The use of two levels of headings and subheadings within the Results section is unclear and too complex. Please use just a single level of headings within the Results.

We have removed the use of subheadings in the results section.

9. The diagram in Fig. 1 is missing an indication that the final number of genes considered after the elastic net penalization is 209.

We hope to guide the reviewer to the bottom right corner of figure 1, where the green text box reads “Elastic net penalised logistic regression model of 209 genes to predict HR status in HGSC”.

10. It is currently impossible to understand the difference between Fig.2B and Fig. 2C without looking at the legend. That is not a good practice, and the authors should consider adding titles and/or subtitles to their plots to avoid this.

Subtitles have been added to figures 2B and 2C, in addition to several others, to enhance legibility.

11. Also, Fig. 2C is mentioned in the text before Fig. 2B, which is also not a good practice. Please reorder the figures accordingly if needed.

We have amended this ordering within the text.

12. Furthermore, some quantitative metric should be used for comparing PCAs in Fig. 2B and 2C beyond the current mention of visual separation.

The PCAs of figure 2B-C are intended to reiterate that the gene expression signal quantified by differential expression analysis can be supported by qualitative observation.

13. The order of panels in Fig. 3 is confusing. Why is panel D at the bottom?

Figure 3 had been arranged to reduce whitespace, we acknowledge that this was not ideal, and have amended the figure so that panels are in alphabetical order.

14. The legend for Fig. 3F is incomplete (the colors for IdentifiHR probability are missing, at least in Mac Preview). Also, these colors seem very similar to the colors used for the Patient HR status. Please update this to avoid issues with the interpretation of the figure.

As above, we apologise for this formatting issue as our manuscript was submitted with completed colour scales. We have regenerated this figure and checked the revised manuscript in the hope of resolving this issue.

15. Similarly, the legend for Fig. 4A is incomplete (missing colors for HRD score).

As above, we apologise for this formatting issue as our manuscript was submitted with completed colour scales. We have regenerated this figure and checked the revised manuscript in the hope of resolving this issue.

16. Supplementary table legends are missing, at least in the version shared for peer review.

Supplementary table legends are provided in the “Contents_supplimentary” sheet of the shared excel file.

17. The mention of supplementary table 6 (“IdentifiHR is an elastic net penalized logistic regression model based on the expression of 209 protein coding genes” section of the Results) is confusing. Shouldn’t this reflect the predictions in the TCGA cohort? Instead, there is an indication of the 209 genes, which should be referenced later in the paragraph.

Supplementary table 6 details the 209 genes used by IdentifiHR, supplementary table 7 details HR status predictions in the TCGA testing cohort. We believe all instances in text are correctly referenced.

18. The installation of the package (at least on a Mac system) generated an error due to the suggested `build_vignettes = TRUE` option. Setting this flag to `FALSE` seems to solve this issue. In addition, it seems that the `SummarizedExperiment` package is also needed and was not included in the default installation option. Although this could be solved by most experienced users, a smoother installation is needed and, preferably, as part of Bioconductor.

We greatly appreciate the reviewer bringing this to our attention and have now corrected the error in our package. We plan to submit the package to Bioconductor for the October release, 2025.

IdentifiHR: predicting homologous recombination deficiency in high-grade serous ovarian carcinoma using gene expression

We thank the reviewers again for their helpful feedback and for the opportunity to improve our manuscript. Our manuscript has now been revised in accordance with the suggested changes. We wish to note that changes or additions to the manuscript text are reflected in red in this response to reviewers. While no major changes were requested, the following minor changes have been included:

- Discussion surrounding the utility of IdentifiHR, and its value with respect to genomic methods of HR status determination, being a HRD score and the CHORD model
- A more comprehensive comparative analysis of CHORD's HR status predictions and those defined by the genomic HRD score
- Commentary surrounding the optimal threshold for HR status stratification used by IdentifiHR
- Updated figure colours

Please find a detailed response to each comments below.

Reviewers' comments:

Reviewer #3 (Remarks to the Author):

I would like to sincerely thank the authors for their comprehensive revision of the manuscript, incorporating the feedback provided by reviewers. Although most of my comments have been successfully addressed, I still have a few additional minor comments regarding my prior requests (I will be referring to them using the previously provided numbering for major/minor comments):

Major comments

3. The authors have now outlined several advantages of IdentifiHR for its usage in HGSC specifically. Considering they have not specifically indicated in their response the changes made to the main manuscript with respect to each of the comments, they should ensure to include these additional details of IdentifiHR advantages in both the Results and the Discussion sections. I strongly recommend that they include the parts of the manuscript changed in a third color in their rebuttal documents for future publications to help peer reviewers.

As requested by the reviewer, we had outlined the advantages of IdentifiHR, with respect to gene expression models and signatures not specifically trained to detect HRD in HGSC. The following text in the manuscript Results and Discussion section address the reviewer's prior comment:

Results: "IdentifiHR had the highest accuracy of any method examined in the TCGA testing cohort, with the HR status of 62 out of 73 samples being correctly predicted (figure 4C). This was followed by MultiscaleHRD with 51, BRCAness with 41 and expHRD with 38 of the 73 samples having correctly predicted HR statuses (figure 4C). There was only one sample that was incorrectly labelled by all four methods. BRCAness and MultiscaleHRD incorrectly labelled a sample each with a germline *BRCA1* mutation. MultiscaleHRD further found 3 of the 6 cases with germline *BRCA2* mutations to be HRP, though one of these did have a HRD score < 42 and was also deemed HRP by IdentifiHR. Further, one case with a somatic *BRCA1* mutation was predicted to be HRP by BRCAness, though had a HRD score > 42. All cases with any potential hypermethylation of the *BRCA1* promoter were labelled as HRD by IdentifiHR, MultiscaleHRD and expHRD, though not BRCAness. However, 7 cases with a loss in heterozygosity at *BRCA1* were HRP, as defined by their HRD score, and 6 of the 7 were labelled as HRD by the BRCAness signature. IdentifiHR predicted 11 of the 13 cases with *BRCA1/2* variants to be HRD. Of the 2 cases labelled as HRP with variants, one had a benign nonsense germline mutation in *BRCA2* (p.K3326*) (49, 50). The other case harboured a somatic *BRCA1* missense mutation (p.C47W) of uncertain functional significance, accompanied by a loss of heterozygosity, raising the possibility of biallelic inactivation. However, the low

HRD score of 27 suggests the variant was unlikely to be pathogenic and that the sample was relatively HR proficient.”

Discussion: “Benchmarking against other gene expression-based signatures to predict HR status, being BRCAness and MultiscaleHRD, in addition to the expHRD model, indicated that IdentifiHR was more accurate in predicting HR status in HGSC. This is in part likely to be because both BRCAness and MultiscaleHRD signatures were optimised to predict HR status in breast cancer, and while expHRD was built for a pan-cancer setting it, like the other methods, is not HGSC specific. IdentifiHR is HGSC-dedicated tool, which is **a non-trivial distinction, especially when considered alongside the incidence of HRD, as it most commonly occurs in HGSC compared to other cancers. Accordingly, IdentifiHR can** capture the unique genomic and transcriptomic features of the disease and has not been tested in alternate cancers.”

Regarding detection thresholds, I appreciate their effort to derive a cohort-specific threshold (although they have not provided the details about how this was done in their response). However, they still did not try my original suggestion to derive two independent thresholds for detection of HRD and HRP (e.g., 75% and 25%), with values in the middle determined as uncertain. I still think this could lead to more clinically meaningful results, although this is a minor point for the overall manuscript. The authors should at least indicate why this is or not a suitable suggestion in their opinion.

We have optimised the probability threshold for HR status, determined by IdentifiHR, for binary classification to prioritise simplicity and reproducibility. To define the optimal threshold, we assessed probability values in intervals of 0.01, to determine a cut-off that would maximise the model’s accuracy. We have clarified this in the Methods section of our manuscript, as follows:

“The probability threshold, at which IdentifiHR predictions are defined as “HRD” was optimised in the TCGA testing cohort **by iterating through probability values from 0 to 1, in intervals of 0.01, and taking the threshold that maximised accuracy.** The AOCS testing cohort provided independent assessment of this adjusted threshold.”

We appreciate the reviewer’s suggestion to optimise two independent thresholds for the classification of HRD and HRP cases, with an intermediate “uncertain” label, given the inherent ambiguity in the genomic HR status of some cases. By providing users with a continuous probability of a sample being HRD alongside analysis of the relationship between this probability and the genomic HRD score, as a representation of the true HR status, we have hoped to support users in making their own choice regarding thresholding. Ultimately, we feel this choice should be left to user discretion as it depends on the context of classification and the related risk tolerance. This is included in the Results section of our manuscript:

“Given this, $P = 0.50$ is used in our package as the default, however, users are provided with the probability of a sample being HRD in the IdentifiHR output and can customise this threshold to re-define discrete HR status, **based on the context of their use and the associated risk tolerance.**”

4-5. Although understanding the authors’ rationale of a gene expression classifier, according to their results CHORD seems to have a better performance than their IdentifiHR approach. Although they focused on the consistency/concordance between methods (Supplementary Figure 8), it’s important to acknowledge that CHORD shows better results at least in the AOCS testing cohort (accuracy 88%, 65/74, vs. 85%, 63/74; according to Supplementary Figure 9E), which is the only place where it was possible to directly review CHORD’s performance. The authors should also include CHORD’s prediction results in the TCGA-OV testing cohort panel in Fig. 4A (as they did in Supplementary Figure 9E), as well as a similar table as the one provided for their IdentifiHR tool in Fig. 3B. The current table in Supplementary Figure 8B compares CHORD vs. IdentifiHR, but not CHORD vs. their ground truth (HRD score derived from SNP array data with a threshold of 42). Also, it is unclear why only 43 samples were included in Supplementary Figure 8B vs. 73 in Figure 3B for the TCGA testing cohort, as all cases in TCGA should have sufficient genomic data for evaluating HR status through CHORD.

To recognise the concordance between HR statuses determined by the HRD score, CHORD and IdentifiHR, we have now included a direct comparison of CHORD and HRD score-defined HR status labels, including a confusion matrix (supplementary figure 8B). The Results section has been updated as follows:

“Across both the TCGA training and testing cohort, the CHORD probability and matched HRD score were positively correlated ($R=0.71$, $p\text{-value} < 2.2 \times 10^{-16}$) and **84.2% concordant** (supplementary figure 8A). **In only the TCGA test cohort, these labels were 86% concordant (supplementary figure 8B).** CHORD labels were 79% concordant with IdentifiHR predictions in the 43 samples of the TCGA testing cohort with WGS, though some variation in the probabilities predicted by each model was observed (supplementary figure 8C-F).”

CHORD could only be used to predict HR status in a subset of the TCGA cohort, as it requires WGS. While there were 73 HGSC cases with both RNA sequencing (required by IdentifiHR) and WES and/or SNP array data (used to define a HRD score), only 43 of these cases had WGS (used by CHORD). Given this missingness, we have avoided adding CHORD predictions directly to figure 4 as we feel it detracts from the overall message of the panel (though we provide the suggested figure below in the interest of directly addressing the comment). We have instead included an oncoplot as supplementary figure 8E examining HR status predictions made in this subset of the TCGA test cohort which have WGS.

Alternate Figure 4A. IdentifiHR outperforms predictive tools of HR status that use gene expression, BRCAness, MultiscaleHRD and ExpHRD, in the TCGA HGSC testing cohort. (A) HRD predictions for each sample (columns) of the TCGA testing cohort. Samples are sorted and colored by HRD score and HR status (top two tracks), and by the predicted CHORD status, where possible. The presence or absence of germline (gBRCA1/2) or somatic (sBRCA1/2) mutations in BRCA1/2 are annotated, with the methylation beta values for the BRCA1 promoter (“Meth BRCA1”), minor allele copy number of BRCA1 (“Minor CN BRCA1”) and the HR status as predicted by IdentifiHR, BRCAness, MultiscaleHRD and ExpHRD are shown. Where the required data was not available, the cell is shaded grey...

Considering the results of CHORD compared to IdentifiHR in the AOCS cohort, as well as the additional evidence requested for TCGA, the authors should include in their discussion a mention that despite the potential usefulness of a gene expression exclusive classifier, current genomic based HR classifiers like CHORD seem to have slightly superior or, at least, equivalent performance.

We appreciate the importance of genomic methods in defining the true HR status of a HGSC sample, and have included the following commentary surrounding their utility to our Discussion section:

“Beyond the HRD score, we show strong concordance between IdentifiHR’s HR status predictions and the CHORD model, which utilises WGS and provides an alternate genomic source of the true HR status. A high degree of concordance was also observed between CHORD and the matched HRD score.”

“CHORD and HRDscore remain the gold standard for HR status determination when genomic sequencing data is available. Where gene expression data is also available, IdentifiHR may be used to complement predictions.”

Minor comments

14. Thanks for revising the color scale for the IdentifiHR color scale. However, the color palette has not been updated to avoid confusion with the patients' HR status and the HRD score in other figures. Considering the different greens used for HRP and HRD status according to IdentifiHR, probably a scale of greens would be better to avoid confusion and be consistent with other figures.

We have now updated the colour scale representing IdentifiHR probabilities to be shades of green.

15. *Now the color scale for the BRCA1 minor allele CN is incorrect (currently displayed as the same color for all copy number values).*

We are unsure why this issue is occurring and apologise for the inconvenience. We have checked the BRCA1 minor allele colour key in our newly submitted figure 4 panel and believe the colour gradient to be correctly presented.